# Linking deeply-sourced volatile emissions to plateau growth dynamics in southeastern Tibetan Plateau

Maoliang Zhang [1✉], Zhengfu Guo[2,3✉], Sheng Xu [1✉], Peter H. Barry [4], Yuji Sano [5,1,12], Lihong Zhang[6], Sæmundur A. Halldórsson [7], Ai-Ti Chen[8], Zhihui Cheng[9], Cong-Qiang Liu[1], Si-Liang Li[1], Yun-Chao Lang[1], Guodong Zheng[10], Zhongping Li[10], Liwu Li[10] & Ying Li[11]

The episodic growth of high-elevation orogenic plateaux is controlled by a series of geodynamic processes. However, determining the underlying mechanisms that drive plateau growth dynamics over geological history and constraining the depths at which growth originates, remains challenging. Here we present He-$CO_2$-$N_2$ systematics of hydrothermal fluids that reveal the existence of a lithospheric-scale fault system in the southeastern Tibetan Plateau, whereby multi-stage plateau growth occurred in the geological past and continues to the present. He isotopes provide unambiguous evidence for the involvement of mantle-scale dynamics in lateral expansion and localized surface uplift of the Tibetan Plateau. The excellent correlation between $^3He/^4He$ values and strain rates, along the strike of Indian indentation into Asia, suggests non-uniform distribution of stresses between the plateau boundary and interior, which modulate southeastward growth of the Tibetan Plateau within the context of India-Asia convergence. Our results demonstrate that deeply-sourced volatile geochemistry can be used to constrain deep dynamic processes involved in orogenic plateau growth.

[1] Institute of Surface-Earth System Science, School of Earth System Science, Tianjin University, Tianjin, China. [2] Key Laboratory of Cenozoic Geology and Environment, Institute of Geology and Geophysics, Chinese Academy of Sciences (CAS), Beijing, China. [3] CAS Center for Excellence in Life and Paleoenvironment, Beijing, China. [4] Marine Chemistry and Geochemistry Department, Woods Hole Oceanographic Institution, Woods Hole, MA, USA. [5] Atmosphere and Ocean Research Institute, The University of Tokyo, Chiba, Japan. [6] School of Geology and Geomatics, Tianjin Chengjian University, Tianjin, China. [7] NordVulk, Institute of Earth Sciences, University of Iceland, Reykjavík, Iceland. [8] Department of Geosciences, National Taiwan University, Taipei, Taiwan, ROC. [9] School of Earth Science and Engineering, Sun Yat-sen University, Guangzhou, China. [10] Northwest Institute of Eco-Environment and Resources, Chinese Academy of Sciences, Lanzhou, China. [11] Institute of Earthquake Forecasting, China Earthquake Administration, Beijing, China. [12] Present address: Center for Advanced Marine Core Research, Kochi University, Kochi, Japan. ✉email: mzhang@tju.edu.cn; zfguo@mail.iggcas.ac.cn; sheng.xu@tju.edu.cn

Orogenic plateaux are characterized by episodic surface uplift and lateral expansion (i.e., upward and outward plateau growth) over geological timescales[1–3]. Several previous studies have investigated how orogenic plateaux attained their present-day elevations and sizes within the context of plate convergence[4,5]. However, current end-member models for the growth of orogenic plateaux remain debated in terms of geodynamic processes that occur at crustal or mantle depths, such as crustal shortening and extrusion[6], inflation of ductile lower crustal flows[7], removal of dense lower lithosphere[8], and whole-mantle convection[9]. Constraining the depths of geodynamic processes involved in orogenic plateau growth is crucial for evaluating the different end-member models. In this respect, geophysical techniques (e.g., seismic tomography[10]) have been successfully used to determine the structure of the crust and mantle beneath orogenic plateaux, from which the geodynamic mechanisms responsible for plateau growth can be explored in combination with geological observations[5] and numerical models[9,11]. Nevertheless, there is still a lack of quantitative proxies available to constrain the depths of plateau growth dynamics.

The orogenic plateau-building processes are accompanied by extensive tectonic and magmatic activity, which can release large amounts of deeply-sourced volatiles (e.g., He, $CO_2$, and $N_2$) into the atmosphere via shallow hydrothermal degassing systems[12,13] (e.g., hot springs). This has been widely recognized in active fault zones and Quaternary volcanic fields of many orogenic plateau regions, such as Anatolia[14], the Central Andes[15,16], Colorado[17–19], and Tibet[20–23]. The relationship between geochemical volatile anomalies in orogenic plateau regions and tectono-magmatic processes can be used to qualitatively assess plateau growth dynamics. Isotopic and elemental compositions of He, $CO_2$, and $N_2$ in the fluids and gases are sensitive tracers of crustal and mantle components involved in the overall volatile inventory[24–26]. In addition, the melting and/or stress-induced dilatancy of deep-seated rocks can lead to detectable changes in volatile geochemistry[27–29], and these changes (e.g., $^3He/^4He$) respond quickly to the tectonic and magmatic processes occurring at depths (e.g., earthquake[30] and volcanic unrest[31]). This suggests that linking geochemical observations at the surface to geodynamic processes at depth is feasible. It is thus conceivable that quantitative estimates of volatile inventories can be used to qualitatively determine the depths of underlying dynamics involved in orogenic plateau growth (i.e., mantle-scale dynamics vs. crustal-scale dynamics).

The Tibetan Plateau is Earth's largest orogenic plateau in a continental collision setting, and has experienced multiple stages of upward and outward growth in the geological past, with the latest stage still ongoing[6,32]. Unlike the southern, northern, and eastern plateau margins that have much sharper topographic gradients, the southeastern Tibetan Plateau (SETP) is characterized by a long-wavelength, low-gradient topography with a gradual decrease in elevation from 4–5 km to 1–2 km over 1000–1500 km (Fig. 1). It is considered one of the most representative margins of orogenic plateaux on Earth that underwent large-scale outward expansion over geological timescales. More importantly, the geodynamic mechanism responsible for the development of the SETP remains fiercely debated, with two competing end-member models: (i) lateral extrusion of rigid blocks along strike-slip faults that may extend into the lithospheric mantle[6], and (ii) southeastward propagation of crustal thickening induced by gravity-driven ductile flows in the lower crust[32]. In addition, more than 800 natural springs have been identified in the SETP, making it an important part of the tectonic and magmatic degassing zone in the Tibetan Plateau[12]. Taken together, we suggest that the SETP provides an excellent natural laboratory to study the plateau-building processes using evidence from volatile geochemistry.

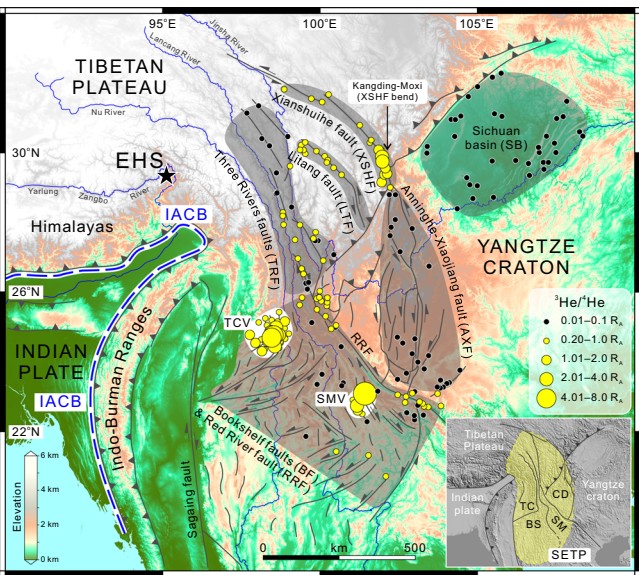

**Fig. 1 Map of the SETP and adjacent region showing simplified tectono-magmatic framework and distribution of air-corrected $^3He/^4He$ values reported relative to air $^3He/^4He$ ($R_A$) for fluids.** Note that air-corrected $^3He/^4He$ values <0.15 $R_A$ are rounded to 0.10 $R_A$, while those ≥0.15 $R_A$ are rounded to 0.20 $R_A$. Yellow shaded area in the inset represents scope of the SETP that consists of several blocks: Tengchong (TC), Baoshan (BS), Simao (SM), and Chuandian (CD). As a result of Indian indentation into Asia and resistance by the Yangtze craton, the SETP is highly active as manifested by large-scale active faulting system and locally distributed Quaternary volcanoes. The active faults (including strike-slip faults, thrusts, and normal faults) are shown as solid gray lines. Full name and abbreviation of each sample group are shown next to the gray shaded area that outlines sample distribution as follows: TRF Three Rivers faults, LTF Litang fault, XSHF Xianshuihe fault, AXF Anninghe-Xiaojiang fault, BF & RRF Bookshelf faults & Red River fault, TCV Tengchong volcanoes, SMV Simao volcanoes, SB Sichuan basin. Note that TCV and SMV are located in the bookshelf faulting zone, and are shown by white shaded areas. The location of India–Asia convergence boundary (IACB), shown as blue dashed line, is defined by southern thrust front of the Himalayas and the thrust front to the west of the Indo-Burman Ranges[6]. The black star symbol denotes the location of eastern Himalayan syntaxis (EHS), which is tectonically important for southeastward growth of the Tibetan Plateau[36]. (The map is generated by Maoliang Zhang using Global Mapper and ArcGIS software).

With the aim of establishing the links between deeply-sourced volatile emissions and plateau growth dynamics, we systematically sampled fluids discharging from the SETP and the adjacent Sichuan basin (Supplementary Fig. 1), and analyzed them for their He–$CO_2$–$N_2$ isotopic characteristics. Ninety-one samples of free gases and water were collected from 51 sites (including natural springs and drilled wells) in the study area. Samples were analyzed for $^3He/^4He$, $δ^{13}C$, $δ^{15}N$, and gas compositions (see "Methods" section and Supplementary Data 1). These newly acquired data were then combined with literature data (Supplementary Data 2) to assess plateau growth dynamics of the SETP. For comparative purposes, samples were divided into the following groups based on their spatial location (Fig. 1): Three Rivers faults (TRF), Litang fault (LTF), Xianshuihe fault (XSHF), Anninghe-Xiaojiang fault (AXF), Bookshelf faults & Red River fault (BF & RRF), Tengchong volcanoes (TCV), Simao volcanoes (SMV), and Sichuan basin (SB). Using He–$CO_2$–$N_2$ data, we show that fluids from several active faults and Quaternary volcanoes have a mantle origin, consistent with lateral expansion and

localized surface uplift in the SETP under the control of mantle dynamics. The fact that $^3$He/$^4$He and total strain rates are positively correlated along the direction of Indian plate motion (i.e., to the northeast) reflects the regional stress field driven by India–Asia convergence, which may have initiated the ongoing stage of plateau growth in the SETP since the mid to late Miocene.

## Results

**Origin of helium**. Helium is chemically inert and $^3$He/$^4$He is a powerful tracer for resolving the origin of deeply-sourced fluids (i.e., identifying mantle vs. crustal inputs)[18,33]. Air-corrected $^3$He/$^4$He values ($R_C/R_A$, where $R_A$ = air $^3$He/$^4$He = 1.39 × 10$^{-6}$) of samples in this study range from 0.01 $R_A$ to 5.39 $R_A$ (Supplementary Data 1; see Supplementary Information for correction method), consistent with $^3$He/$^4$He data (0.01−5.92 $R_A$; Supplementary Data 2) compiled from previous studies. We used

air-corrected $^3$He/$^4$He values to distinguish the relative contributions from mantle and crustal He. The available He isotope data indicate that about 63% of all the sampled sites in the study area have air-corrected $^3$He/$^4$He values >0.20 $R_A$ (~10 times the typical crustal $^3$He/$^4$He value of 0.02 $R_A$[34] and ~2% mantle He). Such enrichments in $^3$He/$^4$He (i.e., >0.20 $R_A$) are considered as unambiguous evidence for the presence of mantle He in the source region[34,35].

Mantle He inputs into the deeply-sourced fluids are prominent in several active fault zones (e.g., XSHF, LTF, TRF, and RRF) and Quaternary volcanic fields (TCV and SMV). In particular, there are three focal regions that are characterized by the presence of significantly high $^3$He/$^4$He values (Figs. 1 and 2a, b): (i) the Kangding-Moxi section of the XSHF ($^3$He/$^4$He = 0.82−3.79 $R_A$; 10−47% mantle He), (ii) the area within ~60 km of Quaternary volcanoes in the Tengchong block ($^3$He/$^4$He = 1.09−5.92 $R_A$; 13−74% mantle He), and (iii) the area within ~40 km of Quaternary volcanoes in the Simao block ($^3$He/$^4$He = 1.04−5.39

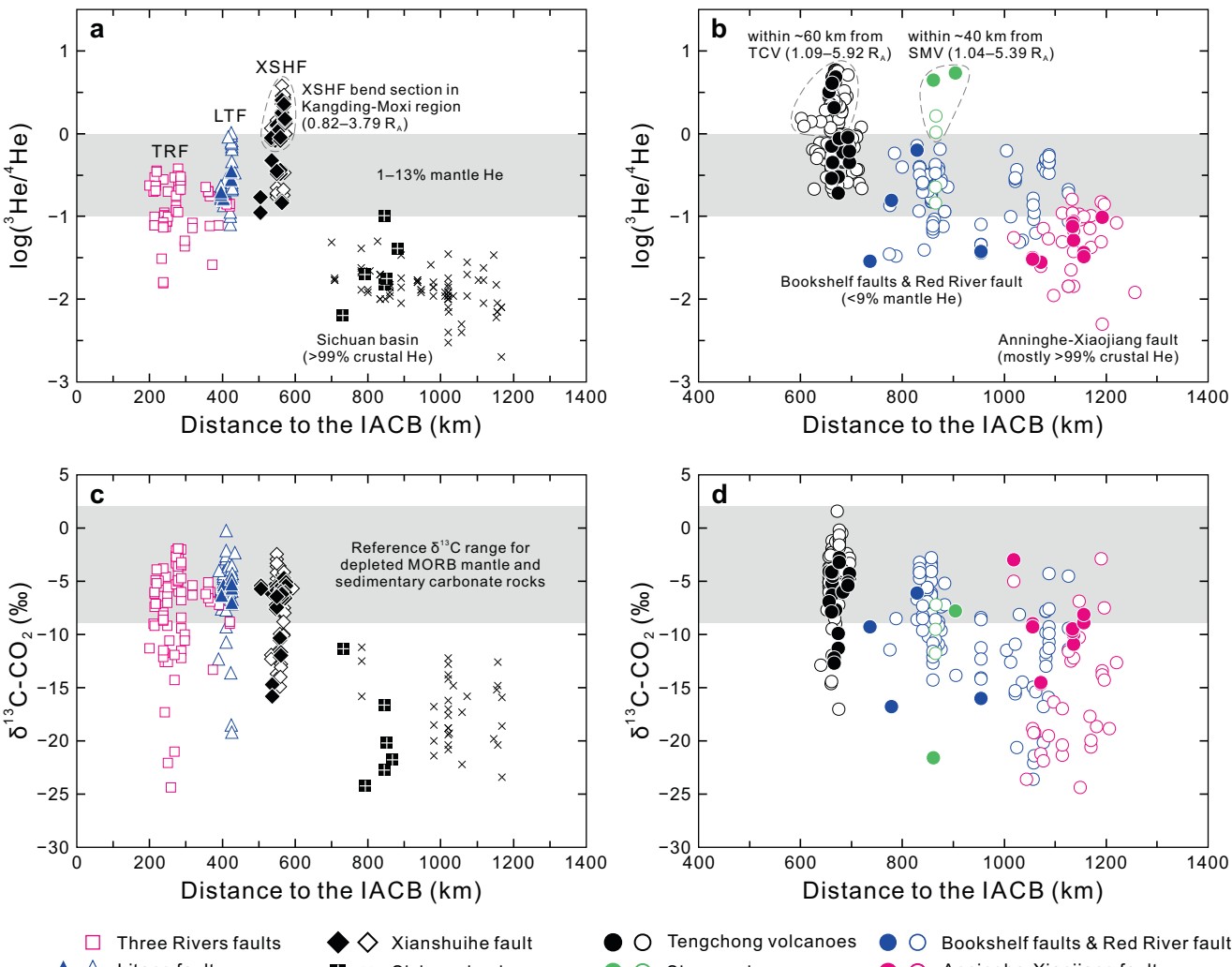

**Fig. 2 Spatial variations of fluid He and C isotopes in the SETP and adjacent region. a** Plot of log($^3$He/$^4$He) versus distance to the IACB (km) for the TRF, LTF, XSHF, and SB samples. **b** Plot of log($^3$He/$^4$He) versus distance to the IACB (km) for the TCV, SMV, AXF, and BF & RRF samples. **c** Plot of δ$^{13}$C-CO$_2$ versus distance to the IACB (km) for the TRF, LTF, XSHF, and SB samples. **d** Plot of δ$^{13}$C-CO$_2$ versus distance to the IACB (km) for the TCV, SMV, AXF, and BF & RRF samples. Abbreviations of sample groups are given in the main text. Filled and open symbols represent samples in this study and literature, respectively. The Sichuan basin samples (black square with white cross—samples in this study; black cross—samples in literature) are shown for comparison with samples of the SETP. Gray shaded areas in **a** and **b** represent 1–13% mantle contributions to He inventory based on crust-mantle mixing calculation (Supplementary Information). Gray shaded areas in **c** and **d** represent the reference δ$^{13}$C range of depleted MORB (mid-ocean ridge basalts) mantle and sedimentary carbonate rocks.

$R_A$; 13–68% mantle He). In stark contrast, the Yangtze craton exhibits typical features of crustal He degassing as represented by the AXF and SB samples with $^3He/^4He$ values <0.10 $R_A$ (i.e., <1% mantle He; Fig. 1). Overall, the spatial variations in mantle He inputs into deeply-sourced fluids appear to closely correspond with active faults and Quaternary volcanoes in the SETP (Fig. 1). Seismic velocity anomalies in the asthenospheric mantle, crustal lithology, and lithospheric thickness cannot explain the observed He degassing patterns, as shown in Supplementary Figs. 2−4.

Spatial variations in air-corrected $^3He/^4He$ values are further constrained by the plot of log($^3He/^4He$) versus sample distance to a tectonic boundary (Fig. 2), which is defined as the India–Asia convergence boundary (IACB; see details in "Methods" section). Figure 2a illustrates the features of deeply-sourced He degassing along the direction of the Indian plate motion to the northeast. Specifically, the TRF and LTF samples have average $^3He/^4He$ values of 0.16 ± 0.09 $R_A$ (1σ, n = 28) and 0.35 ± 0.25 $R_A$ (1σ, n = 13), respectively, while the $^3He/^4He$ values of the XSHF samples are averaged at 1.02 ± 0.76 $R_A$ (1σ, n = 20). This defines an increasing $^3He/^4He$ trend with increasing distance to the IACB near eastern Himalayan syntaxis (EHS; Figs. 1 and 2a). In particular, the $^3He/^4He$ values appear to culminate in bend section of the XSHF in the Kangding-Moxi region ($^3He/^4He$ = 1.52 ± 0.58 $R_A$, n = 12; Fig. 2a). We calculated the Pearson correlation coefficient r = 0.737 with 95% BCa confidence interval (CI) [0.600, 0.840], p < 0.001 (n = 53) for log($^3He/^4He$) and distance to the IACB of the samples from the TRF, LTF, and XSHF bend. It should be noted that such positive correlation can only be observed in the three sample groups along the strike of Indian indentation into Asia (i.e., the TRF, LTF, and XSHF). Further to the northeast, a sudden drawdown in $^3He/^4He$ is observed between the XSHF and the tectonically stable Sichuan basin, where the fluids have average $^3He/^4He$ value of 0.020 ± 0.018 $R_A$ (1σ, n = 35) and >99% crustal He contributions (Fig. 2a).

In Fig. 2b, the $^3He/^4He$ values of the AXF, BF & RRF, TCV, and SMV samples are plotted against sample distance to the IACB west of the Indo-Burman Ranges. This plot highlights the He degassing features of the southern SETP, which has experienced nearly E–W lithospheric extension in the late Cenozoic[6,36]. With the exception of the sites within about 40–60 km of Quaternary volcanoes (i.e., TCV and SMV), fluids discharging from non-volcanic regions (i.e., active fault zones) of the southern SETP generally have $^3He/^4He$ values lower than 1 $R_A$ (Figs. 1 and 2b). The BF & RRF samples have average $^3He/^4He$ values of 0.25 ± 0.18 $R_A$ (1σ, n = 42), with as high as 9% mantle He contributions, which are generally consistent with the observed He isotope signatures in TRF and LTF samples (Fig. 2b). In contrast, the AXF samples ($^3He/^4He$ = 0.07 ± 0.04 $R_A$ (1σ, n = 26)) are dominated by crustal He degassing. Taken together with the SB samples, we show that He is mostly derived from the crust throughout the Yangtze craton (Fig. 1), irrespective of tectonic features (i.e., tectonically stable sedimentary basin versus activated faults in response to southeastward growth of the Tibetan Plateau).

**Origin of major volatiles**. The $\delta^{13}C$ values (versus Vienna Pee Dee Belemnite; VPDB) of gas-phase $CO_2$ samples range from −24.2‰ to −2.8‰ (Supplementary Data 1). $CO_2$ released from the tectonically active SETP has relatively high $\delta^{13}C$ values (Fig. 2c, d), as observed in active fault zones (e.g., TRF, LTF, XSHF, and RRF) and Quaternary volcanic fields (e.g., TCV). In contrast, samples from the Yangtze craton generally have lower $\delta^{13}C$–$CO_2$ values compared to reference $\delta^{13}C$ range of the depleted mantle (−6.5 ± 2.5‰[37]) and sedimentary carbonate rocks (0 ± 2‰[38]).

This is shown by the SB samples with $\delta^{13}C$-$CO_2$ ranging from −24.2‰ to −11.2‰, suggesting an organic origin of $CO_2$ possibly related with thermogenic decomposition of organic matter. For each sample group, the $\delta^{13}C$–$CO_2$ and $CO_2/^3He$ values exhibit large variations, which are attributable to mixing of multiple end-member components and elemental/isotopic fractionation caused by solubility-controlled phase separation processes (e.g., the preferential partition of He into vapor phase relative to $CO_2$ during hydrothermal degassing[39–41]; Supplementary Figs. 5 and 6).

Using a ternary mixing model originally proposed by Sano and Marty[24], we calculated the proportions of mantle (MORB), carbonate (CAR), and organic matter (ORG) (Fig. 3a). The AXF and SB samples are not considered for this calculation, because the fluids from the Yangtze craton generally have an assimilated crustal origin as indicated by generally low He isotopes. The results show that mantle-derived $CO_2$ is being released from the XSHF, LTF, TRF, and BF & RRF, although the proportions of MORB mantle are variable (8.1% ± 13.5% (1σ, n = 42)) as a result of elemental/isotopic fractionation of the He–$CO_2$ systematics. However, there are no obvious spatial variations in mantle $CO_2$ contributions between different active fault zones. For Quaternary volcanic fields, the fluids appear to have higher mantle $CO_2$ proportions (mean = 23.4% ± 24.7% (1σ, n = 18) for TCV, and 27.8–62.8% (n = 3) for SMV) relative to the fluids from non-volcanic regions. This may reflect the efficient mantle $CO_2$ transport in terms of degassing of residual magma bodies at crustal levels beneath Quaternary volcanoes[42,43]. It should be noted that the $\delta^{13}C$–$CO_2$ and $CO_2/^3He$ values of several samples are likely modified by secondary fractionation processes (Fig. 3a; e.g., calcite precipitation[44]). However, the interpretation of release of mantle $CO_2$ appears to be robust for most tectonically and volcanically active regions in the SETP.

Nitrogen isotopes ($\delta^{15}N$ (versus air) of $N_2$) were determined for 34 samples in this study, yielding a $\delta^{15}N$–$N_2$ range of −2.1‰ to +5.8‰ (Supplementary Data 1). Following a model[25] based on mixing of the mantle (MORB), sediments (SED), and air (AIR) end-members (Fig. 3b), we calculated the $N_2$ inventory to evaluate the possibility of mantle $N_2$ emissions in the SETP. Several samples appear to be affected by elemental fractionation between He and $N_2$ during vapor-melt separation[45]. Most $N_2$ in the fluids is derived from air or air-saturated water (air-derived $N_2$ proportion = 71.4 ± 23.2% (1σ, n = 19)), which is commonly observed in thermal springs. Correspondingly, the deeply-sourced $N_2$ takes up <30% of the total $N_2$ inventory on average. It is noted that mantle $N_2$ emissions are identified in several active fault zones (e.g., XSHF) and Quaternary volcanic fields (e.g., TCV). In particular, mantle $N_2$ contributes 8.4 ± 8.0% (1σ, n = 10) of the total $N_2$ released by active faults in non-volcanic regions of the SETP.

**Linking deeply-sourced He degassing to regional stress field**. As discussed above, the He–$CO_2$–$N_2$ systematics of hydrothermal fluids suggest the release of mantle volatiles from active fault zones and Quaternary volcanoes in the SETP. Given that He isotopes are chemically inert and vary by over two orders of magnitude between the mantle and crustal end-members, air-corrected $^3He/^4He$ values are more reliable than $\delta^{13}C$ and $\delta^{15}N$ in identifying mantle- and crust-derived components. The $^3He/^4He$ values observed in the surface fluids are mainly controlled by two processes: (i) release (via dilatancy/melting of the crustal and/or mantle rocks[29,42]) from the minerals in which He is produced or trapped, and (ii) transport from the source to the surface during which radiogenic $^4He$ can be assimilated into the uprising fluids[34,42]. In this respect, fault permeability is expected to play an important role in the He inventory by influencing (i) the

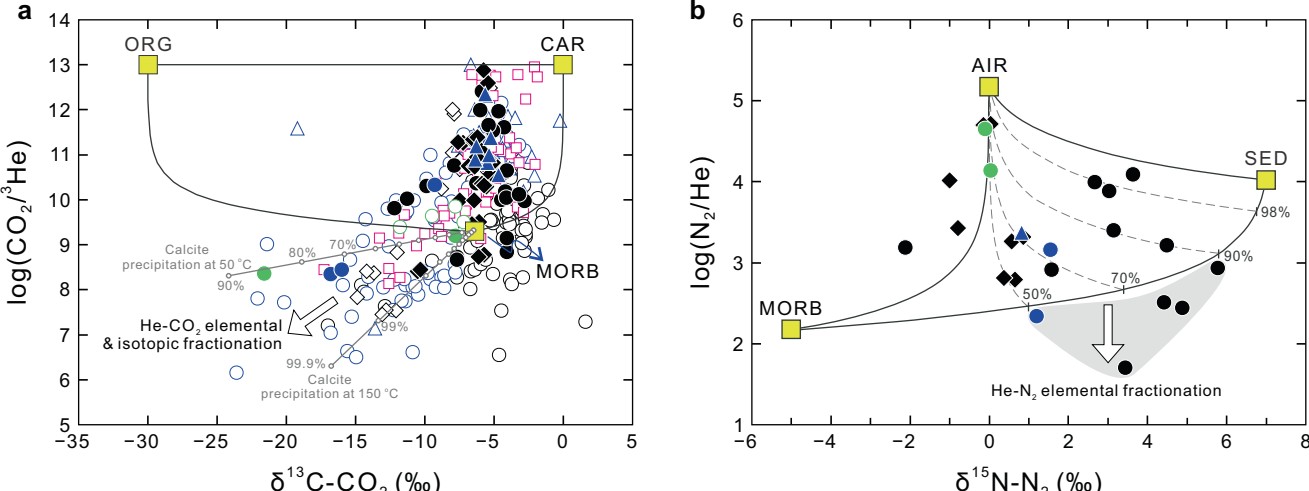

**Fig. 3 Ternary mixing models showing origin of CO$_2$ and N$_2$ in hydrothermal fluids. a** Plot of log(CO$_2$/³He) versus δ¹³C–CO$_2$ (‰). MORB depleted mantle, CAR carbonate, ORG organic matter. Sample symbols are as in Fig. 2. The effect of elemental and isotopic fractionation between He and CO$_2$ is shown by the white arrow. Solid gray lines with white-filled dots represent predicted calcite fractionation model trends (shown as % CO$_2$ loss; see ref. [44]) at a temperature of 50 °C and 150 °C, respectively, assuming a mantle-like starting composition of δ¹³C = −6.5‰ and CO$_2$/³He = 2 × 10⁹. Samples from the AXF are not considered for this calculation, because they have a pure crustal origin. **b** Plot of log(N$_2$/He) versus δ¹⁵N–N$_2$ (‰). MORB depleted mantle, SED sediments, AIR air. Numbers (50, 70, 90, and 98%) on the MORB–SED mixing curve represent the proportions of sediments. Several samples are affected by elemental fractionation between He and N$_2$ as a result of their solubility differences during vapor separation. Further details on the results of ternary mixing calculations given in the main text.

transport time of mantle He in the crust, and thus (ii) the extent to which mantle He is contaminated by crustal He. A previous study[26] undertaken in the Basin and Range Province, western North America, revealed that high ³He/⁴He values occur where the extension and shear strain rates are high. This suggests that deformation-enhanced fault permeability is able to facilitate the release of mantle He in non-volcanic regions. Here, we compare the distribution of ³He/⁴He values with geodetic strain rates of the SETP to evaluate the relationship between deeply-sourced He degassing and regional stress field (Fig. 4). The TCV and SMV samples are not considered for this comparison, because the transfer of mantle He to the surface is most efficient by melting of the mantle and melt intrusion into the crust[42]. In this case, mantle He degassing from Quaternary volcanoes may be less dependent on fault permeability than that in non-volcanic regions, where shallow crustal magma chambers are absent.

We compiled strain rate data of the SETP and adjacent region from the Global Strain Rate Model v2.1 (see ref. [46]), from which average strain rates of the studied active faults were calculated based on spatial distribution of the fault system (Supplementary Information). Figure 4a shows the distribution of total strain rates[46] and active faults in the SETP and adjacent region. High strain rates are observed along the XSHF (123 ± 60 nstrain/yr (1σ, n = 149)) and AXF (58 ± 29 nstrain/yr (1σ, n = 576)), which collectively form a nearly NW–SE trending plateau boundary fault characterized by high shear strain rates[47] (Fig. 4a). In contrast, the interior faults of the SETP are characterized by relatively low strain rates, as shown by the TRF (17 ± 21 nstrain/yr (1σ, n = 595)), LTF (30 ± 12 nstrain/yr (1σ, n = 85)), and BF & RRF (30 ± 15 nstrain/yr (1σ, n = 1398)). A prominent feature of the geodetic strain rate field is that the strain rates generally increase from the TRF, via the LTF, to the XSHF (Fig. 4a), consistent with northeastward motion of the Indian plate (i.e., the direction of India–Asia convergence). It is possible that the stresses driven by India–Asia convergence are high and compressive within a narrow zone of the plateau boundary fault (especially the localized bend section of the XSHF). In contrast,

the stresses across the interior faults of the SETP are relatively low and absorbed over a much broader region. This agrees well with the observation that distributed deformation characterizes the plateau interior[48] and localized deformation occurs along the plateau boundary[49]. Although the magnitude of strain rate is also influenced by lithospheric rheology, the contrasting geological and geodetic slip rates between boundary and interior faults of the SETP (see details in "Discussion" section) suggest a fundamental role of regional tectonic stresses in determining the strain rate field. High strain rates are also observed along the Sagaing fault (Fig. 4a) and the IACB (e.g., the Himalayan fold-and-thrust belt[50]; not shown in Fig. 4), which are beyond the scope of this more focused geochemical study.

Plotting ³He/⁴He values against strain rates allows the relationships between deeply-sourced He degassing and regional stress field to be constrained. Spatially, the variable levels of ³He/⁴He values exhibit a moderate correlation with total strain rates of the sampling sites in all active fault zones of the SETP, for which the Pearson correlation coefficient has been calculated (r = 0.483 with 95% BCa CI [0.321, 0.628], p < 0.001 (n = 129)). Following the direction of the Indian plate motion to the northeast (Fig. 4a), we identified strong correlations between the variables of ³He/⁴He, total strain rate, and distance to the IACB near the EHS (Fig. 4b, c). Based on compiled data from the TRF, LTF, and XSHF bend (Supplementary Data 3), we calculated a Pearson's coefficient of r = 0.752 with 95% BCa CI [0.689, 0.857], p < 0.001 (n = 53) for ³He/⁴He and total strain rate (Fig. 4b), and r = 0.810 with 95% BCa CI [0.717, 0.876], p < 0.001 (n = 53) for distance to the IACB and total strain rate (Fig. 4c). It is thus straightforward that both ³He/⁴He values and total strain rates increase with increasing distance to the IACB near the EHS (Figs. 2a and 4). Unlike the above observation, the AXF is characterized by moderately high total strain rates but low ³He/⁴He values (mostly < 0.10 $R_A$; Fig. 2b), suggesting that crustal He degassing dominates such cratonic regions. If AXF sites are not considered, the Pearson's coefficient r would be 0.583 with 95% BCa CI [0.423, 0.723],

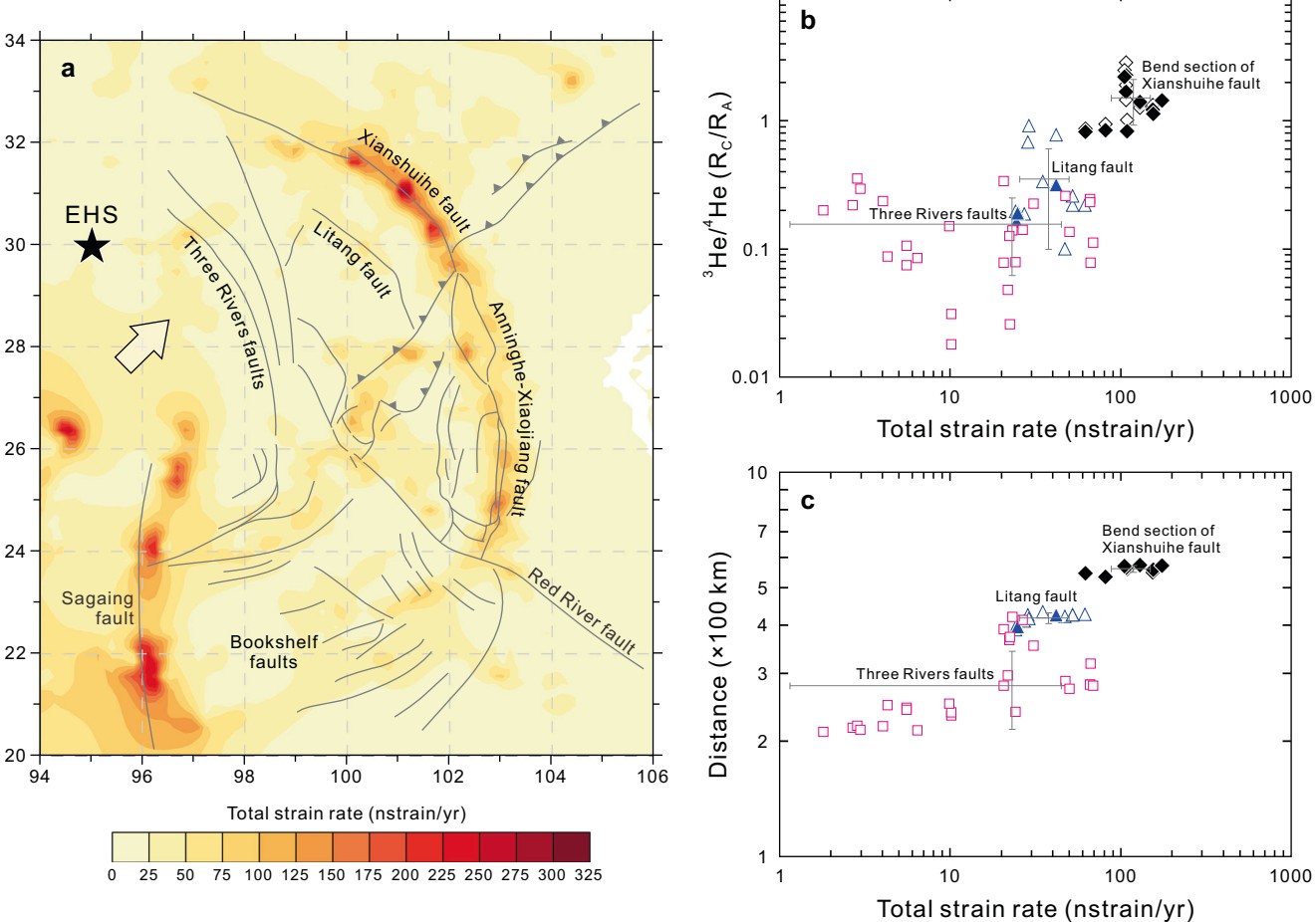

**Fig. 4 Comparison between $^{3}$He/$^{4}$He distribution and geodetic strain rate field along the direction of Indian indentation into Asia. a** Contour map showing total strain rates of the SETP and adjacent region based on the Global Strain Rate Model[46]. Major active faults and the EHS (eastern Himalayan syntaxis) are shown as in Fig. 1. The white arrow represents the direction of Indian plate motion to the northeast. **b** Plot of $^{3}$He/$^{4}$He ($R_{C}/R_{A}$) versus total strain rate (nstrain/yr) showing positive correlation starting from the TRF ($^{3}$He/$^{4}$He = 0.16 ± 0.09 $R_{A}$ (1σ, n = 28); total strain rate = 23.1 ± 21.9 nstrain/yr (1σ, n = 28)), via the LTF ($^{3}$He/$^{4}$He = 0.35 ± 0.25 $R_{A}$ (1σ, n = 13); total strain rate = 37.7 ± 12.2 nstrain/yr (1σ, n = 13)), to bend section of the XSHF ($^{3}$He/$^{4}$He = 1.52 ± 0.58 $R_{A}$ (1σ, n = 12); total strain rate = 118.9 ± 30.6 nstrain/yr (1σ, n = 12)). Average $^{3}$He/$^{4}$He value and average strain rate of each sample group are shown in 1σ level (i.e., solid gray lines). **c** Plot of distance to the IACB (×100 km) versus total strain rate (nstrain/yr) showing increasing total strain rates with increasing distance to the IACB, as observed from the TRF (distance = 278 ± 64 km (1σ, n = 28)), via the LTF (distance = 417 ± 13 km (1σ, n = 13)), to bend section of the XSHF (distance = 560 ± 12 km (1σ, n = 12)). Average distance to the IACB and average strain rate of each sample group are shown in 1σ level (i.e., solid gray lines).

$p < 0.001$ ($n = 103$) for $^{3}$He/$^{4}$He values and total strain rates of active fault zones without cratonic affinities. This suggests the potential relationships between deeply-sourced He degassing and regional stress field, which influences the fault permeability in terms of strain partitioning across active faults.

## Discussion

Southeastward growth of the Tibetan Plateau is expressed by two first-order tectonic features: (i) outward expansion along major strike-slip faults[6,36], and (ii) clockwise block rotation around the EHS[51]. The identification of mantle He degassing along the XSHF, LTF, TRF, and RRF (Fig. 1) reveals the existence of a lithospheric-scale strike-slip fault system in the SETP that allows the release of mantle He. This provides solid evidence for constraining the depths of major strike-slip faults that may extend into the lithospheric mantle[6]. Our $^{3}$He/$^{4}$He results indicate that the clockwise strike-slip processes around the EHS (i.e., lateral expansion of the SETP) are occurring at the mantle scale, rather than being limited to shallower (i.e., crustal-only) depths. It should be noted that constraining the surface uplift history of the

SETP is not possible using geochemical data alone, and requires a multidisciplinary approach (e.g., geodynamical, geophysical, and geochemical observations). The fact that we observe high $^{3}$He/$^{4}$He values in the fluids of regions that were rapidly uplifted lends credence to this approach. Located in Kangding-Moxi section of the XSHF, the Gongga Shan massif has risen more than 3000 m over mean elevation of the plateau margin[52] since the beginning of its rapid exhumation and uplift in the mid to late Miocene[49]. We suggest that the co-occurrence of high $^{3}$He/$^{4}$He values, high strain rates, and rapid surface uplift in the Kangding-Moxi region indicates mantle-wide-scale dynamic processes occurring beneath bend section of the XSHF. This may have resulted in the formation of a transpressional faulting system[53] with fault splays extending into the mantle and localized surface uplift along the XSHF bend. Considering that weak crustal zones associated with radiogenic heating in a thickened crust[7] are expected to release large amounts of radiogenic He into deeply-sourced fluids, the lower crustal flow model of plateau growth[7] is at odds with observed high $^{3}$He/$^{4}$He values. Therefore, significant proportions of mantle He that is degassing along lithospheric-scale faults

would strongly suggest involvement of mantle dynamics in southeastward growth of the Tibetan Plateau.

Based on comparison between $^3He/^4He$ distribution and geodetic strain rate field, we suggest that deeply-sourced He degassing patterns in the SETP may present a "snapshot" of regional stress field that dominates the ongoing plateau growth. In particular, the well-correlated $^3He/^4He$ values and strain rates from the TRF to the XSHF bend (Fig. 4b) can reflect the control of India–Asia convergence on the non-uniform distribution of tectonic stresses in the northern SETP and the deformation-related variations in fault permeability. The XSHF, especially its bend section, is a focal zone of high compressive stresses and has enhanced fault permeability as a result of the India–Asia convergence and resistance by the Yangtze craton. In contrast, the stresses exerted on the TRF and LTF are relatively lower as implied by geodetic measurements[46–48] and $^3He/^4He$ results (Fig. 4b). Unlike the XSHF characterized by high compressive shear strain[47], the southern SETP is dominated by extensional stresses as revealed by geological[6] and geodetic[47] evidence. By comparing $^3He/^4He$ values and total strain rates between the XSHF and non-volcanic regions in the southern SETP (e.g., BF), we suggest that compressive shear stresses are more favorable for (i) fault splaying into the depth, (ii) enhancement of fault permeability, and thus (iii) rapid release of mantle He, than the case of extensional stresses. This is consistent with the observation in non-volcanic regions of northern Basin and Range Province[26]. Overall, the $^3He/^4He$ distribution agrees well with regional stress field of the SETP, which drives the plateau growth within the context of the India–Asia convergence.

As discussed above, the links between deeply-sourced volatile emissions and plateau growth dynamics have been established for the present-day SETP, which is important for understanding how India–Asia convergence drives the ongoing growth of the SETP and the outgassing of deeply-sourced volatiles. It is also intriguing how (i) the current stage of lateral expansion and localized surface uplift initiated in the geological past, and (ii) the deeply-sourced volatile emissions responded to plateau growth. Considering that the outward and upward growth of the Tibetan Plateau are ongoing along active faults, the fault initiation/reactivation ages are plausible to represent onset of the currently ongoing stage of plateau growth[6,54]. The active faults in the SETP generally initiated or reactivated in the mid to late Miocene (e.g., 13–10 million years ago (Ma) in the case of the XSHF; Supplementary Data 4), consistent with the timing of fault displacement reversal in the southern SETP[55]. Moreover, late Cenozoic mantle-derived volcanism (ca. 14 Ma to the present; Supplementary Fig. 1 and Data 5) also occurred in the SETP and adjacent region, including Quaternary volcanoes in the Tengchong and Simao blocks. Therefore, the mid to late Miocene is likely a geodynamically important period for the beginning of outward and localized upward growth in the SETP.

We have shown that deeply-sourced He degassing from active fault zones is largely influenced by modern regional stress field of the SETP (Fig. 4). To understand how the deeply-sourced volatile emissions responded to plateau growth processes, the regional stress field during the past ~13–10 Ma should be reconstructed for the SETP in future investigations. Nevertheless, it may be plausible to evaluate the stresses exerted on faults over geological timescales using observations such as exhumation rate and geological slip rate. A recent study on the Gongga Shan batholith revealed rapid exhumation and uplift since ~9 Ma in the Kangding-Moxi region[49], which is suggested to be coeval with the onset of strike-slip motion along the XSHF. It is possible that the Kangding-Moxi region has been dominated by high compressive stresses since the mid to late Miocene, which facilitated bending of the fault system and related transpressional uplift[53]. Given that high $^3He/^4He$ values, high total

strain rates, and rapid surface uplift are collectively observed in the Kangding-Moxi region, we suggest that significant mantle He degassing may have occurred in the bend section of the XSHF since its initiation ~13–10 Ma. Additionally, the geological slip rate of the XSHF (e.g., 6.8–7.6 mm/yr since ~9 Ma[49]) is likely higher than that of the plateau interior faults (e.g., the LTF with slip rate of 0.9–2.3 mm/yr since ~6 Ma[56]), consistent with their differences in geodetic slip rates[57] at present. The contrasting slip rates over geological timescales suggest that the stresses exerted on the interior faults of the SETP are lower than those of the plateau boundary fault. In this case, it is possible that mantle He degassing along the plateau interior faults is less significant than degassing of the XSHF bend since mid to late Miocene time.

We conclude that deeply-sourced volatile emissions may have responded quickly to onset of the ongoing plateau growth. Geodynamically, a reorganization of regional stress field since the mid to late Miocene is required to trigger lateral expansion and localized surface uplift of the SETP. Possible causes for regional stress reorganization and tectonic reactivation in the SETP may be the shift in direction of the India–Asia convergence from ~N30°E to ~N10°E in the mid to late Miocene[58] (ca. 13–10 Ma). As proposed in previous studies[59,60], the rollback of subducted Indian oceanic slab and trench retreat are consistent with geological observations of the SETP and adjacent region[59], and are taken as driving mechanisms for growth of the SETP in numerical models[60]. If the Indian slab rollback and trench retreat occurred in mid to late Miocene time, it would be expected to observe the nearly E–W extension and the reversal of fault displacements[55] in the overriding plate (i.e., the southern SETP). This would facilitate onset of the strike-slip motion around the EHS and thus initiate lateral expansion of the SETP (Fig. 5a). As shown in Fig. 5b, the near-field response to slip transfer at the confluence of a major strike-slip fault and a subparallel strike-slip fault[61] can account for transpressional uplift of the Gongga Shan massif since mid to late Miocene time (i.e., localized surface uplift).

It should be noted that the direction and rate of plateau growth in the SETP cannot be deduced from He–$CO_2$–$N_2$ systematics of hydrothermal fluids alone, which requires a multidisciplinary approach. Nevertheless, our geochemical findings can be used to (i) constrain the depths of plateau growth dynamics, and (ii) evaluate how the India–Asia convergence dominates the strain rate distribution in the SETP. This provides important insight into the deep dynamic processes that contributed to shaping Earth's surface over geological timescales. Our results present a good correlation between deeply-sourced He degassing patterns and regional stress field, which links volatile geochemistry to plateau growth dynamics in the SETP. For the plate convergence settings worldwide (especially orogenic plateau regions), the relationships between volatile geochemistry and regional stress field should be explored based on careful evaluation of related controlling factors for volatile geochemical anomalies (as stated in this study for $^3He/^4He$, $\delta^{13}C$, and $\delta^{15}N$ values), including mantle seismic velocity structure, proximity to Quaternary volcanoes, crustal lithology, and physico-chemical conditions of hydrothermal degassing systems that can affect elemental and isotopic fractionation of the He–$CO_2$–$N_2$ systematics[35,39,44,45].

## Methods

**Field campaigns and sample collection**. Free gas and water samples were collected from the SETP and adjacent region during six field campaigns carried out in 2011, 2018, 2019, and 2020, respectively. Our samples cover a broad region (Supplementary Fig. 1) and represent the fluids discharging from different tectonic units of the study area, including Quaternary volcanic fields, (non-volcanic) active faults, and sedimentary basin in the tectonically stable Yangtze craton. Low-He diffusivity glass containers were used to collect free gases from natural springs following water displacement method. We used copper tubes and glass containers to collect water samples from drilled wells in the Sichuan basin (Supplementary

**a**

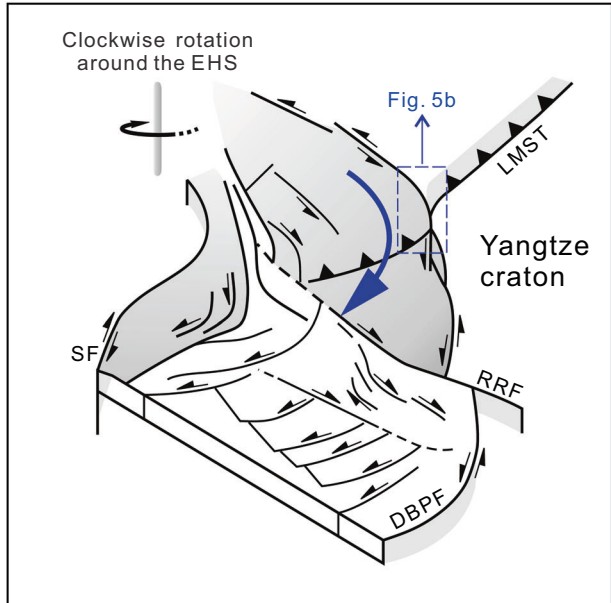

**b**

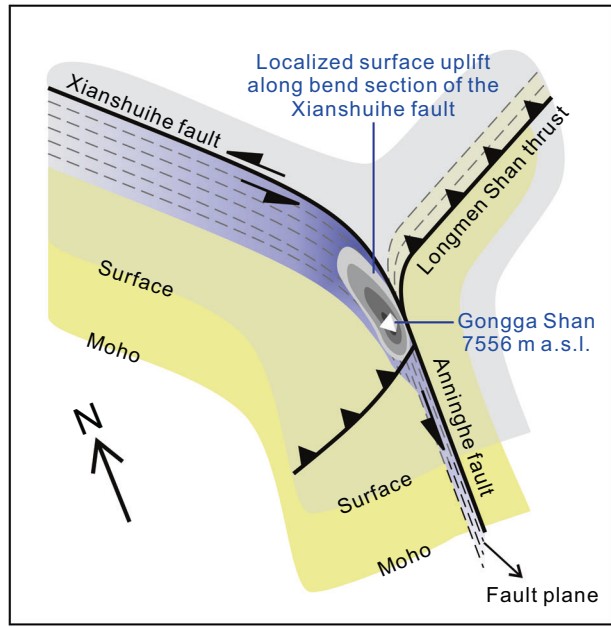

**Fig. 5 Schematic model showing 3D sketches of the structural setting of the SETP and the styles of outward and localized upward growth of the Tibetan Plateau. a** Lateral expansion of the Tibetan Plateau via clockwise strike-slip motion of the blocks around the eastern Himalayan syntaxis (EHS) (as shown by the blue arrow; modified from ref. [66]). RRF Red River fault, DBPF Dien Bien Phu fault, SF Sagaing fault, LMST Longmen Shan thrust. A regional stress reorganization is required for initiation of the lateral expansion stage that continues to the present (see details in the main text). **b** Localized surface uplift in the Kangding-Moxi region that possibly resulted from transpressional tectonism in bend section of the Xianshuihe fault[53]. Yellow shaded area represents the Moho interface, while the gray shaded (and translucent) area represents the local surface. Fault planes of the Xianshuihe fault (left-lateral strike-slip), Anninghe fault (left-lateral strike-slip), and Longmen Shan thrust are shown by the black dashed lines with 10 km intervals (not to scale), based on the styles of active faulting and local crustal thickness[67]. The bend section of the Xianshuihe fault is characterized by high compressive stresses (shown in fading blue), which have contributed to localized surface uplift (as represented by the concentric contours filled in light gray to dark gray) including the rise of the Gongga Shan massif (7556 m above sea level).

Information) and natural springs without visible bubbling gases. Robust sampling procedures were adopted to minimize the influence from air contamination and samples were analyzed as soon as possible after the fieldwork. Further details on the field campaigns, sampling procedures and sample distribution are outlined in Supplementary Information.

**Laboratory analyses**. Gas compositions, $^3He/^4He$, $\delta^{13}C–CO_2$, $\delta^{13}C–CH_4$, and $\delta^{15}N–N_2$ were measured for selected gas and water samples (Supplementary Data 1) at Scripps Institution of Oceanography (SIO), University of California San Diego, Lanzhou Center for Oil and Gas Resources (LCOGR), Chinese Academy of Sciences, and Atmosphere and Ocean Research Institute (AORI), The University of Tokyo, respectively. Data from different laboratories are in good agreement, as shown by the measured $^3He/^4He$ values of samples from the same site (e.g., 2011 sample Teng-11-26 vs. 2019 sample YN190502-2, and 2018 sample EDQ01 vs. 2020 sample EDQ2001; Supplementary Data 1).

The 2011 samples ($n = 30$) were analyzed for He, Ne, $CO_2$, and $N_2$ elemental and isotopic compositions at SIO, University of California San Diego. Samples were released into an ultra-high vacuum (UHV) stainless steel purification line to separate the non-condensable gases (He, Ne, and $N_2$) and condensable gases ($H_2O$ and $CO_2$). The light noble gases (He and Ne) were then isolated from active gases ($N_2$, CO, and $CH_4$) and heavy noble gases (Ar, Kr, and Xe) by exposure to an activated charcoal trap at $-196$ °C, a 700 °C hot Ti-getter, and a cryogenic trap lined with activated charcoal (held at $<20$ K prior to releasing He into a MAP-215 noble gas mass spectrometer). $^3He/^4He$ and $^4He/^{20}Ne$ ratios were measured in static mode and calibrated against standard aliquots of air run at least twice a day under identical conditions[62]. Total amount of $CO_2$ was measured using a capacitance manometer in a calibrated volume, and was then combined with the mass spectrometer-derived $^3He/^4He$ value and He abundance to calculate the $CO_2/^3He$ ratio. The $CO_2$ aliquot was transferred to a Thermo Finnigan Delta XP Plus Isotope Ratio Mass Spectrometer (IRMS) for analyses of carbon isotope values, which are reported in $\delta^{13}C$ relative to the international reference standard Vienna Pee Dee Belemnite (VPDB) and have uncertainties of $<0.1‰$ based on repeat analyses of standards and samples[39]. The abundance and isotopic compositions of $N_2$ were analyzed by a modified VG5440 mass spectrometer. Average $\delta^{15}N$ (reported relative to air) reproducibility of the Scripps-pier air standard value is $\pm0.48‰$ ($1\sigma$)[63].

The 2018 samples ($n = 44$) were analyzed for chemical compositions and He–C–N isotopes at LCOGR, Chinese Academy of Sciences. Compositions of free gases (e.g., $CO_2$, $CH_4$, $O_2$, $H_2$, $N_2$, He, and Ar) were measured by a MAT 271 mass spectrometer. About ~1 ml of gases was extracted from sample bottle using a syringe and then injected to the sample entrance line connected to the mass spectrometer. Repeated analyses of air standard yield analytical error $<2\%$ for major gas species ($N_2$, $O_2$, Ar, and $CO_2$). The $\delta^{13}C–CO_2$ and $\delta^{15}N–N_2$ values were analyzed by a Thermo Finnigan Delta XP Plus Isotope Ratio Mass Spectrometer (IRMS). The average reproducibility is better than $\pm0.5‰$ for interlaboratory standards[64]. Before measurement of He and Ne isotopes, the free gases were purified using a spongy titanium furnace at 800 °C and a Zr–Al getter running at room temperature to remove active gases ($H_2O$, $O_2$, $N_2$, and $CO_2$) and $H_2$, respectively. The abundance and isotopic compositions of He and Ne were then analyzed by a Noblesse noble gas mass spectrometer[64]. The air collected from the top of Gaolan Mountain was analyzed to meet laboratory standards ($^3He/^4He =$ $1.4 \times 10^{-6}$) during routine measurement. Analytical error of $^3He/^4He$ value is less than $\pm1.5\%$.

The 2019 and 2020 samples ($n = 17$) were analyzed for chemical compositions and He-C isotopic compositions at AORI, The University of Tokyo. A small portion of gas sample was introduced into a quadrupole mass spectrometer (Prisma QMS200) through a variable leak valve to measure chemical compositions by comparing peak heights of the sample with those of standard gases. After extraction of gases from the water phase to a vacuumed lead glass container, $^3He/$ $^4He$ and $^4He/^{20}Ne$ ratios of the gases was measured by a high-resolution Helix SFT noble gas mass spectrometer[65]. Before analyses of He and Ne abundances and isotopes, a portion of gas samples was introduced into an all-metal high-vacuum line, where He and Ne were purified using hot Ti getters and charcoal traps at liquid nitrogen temperature. $^4He/^{20}Ne$ ratios were measured using a QMS on-line and He was separated from Ne at 40 K by a cryogenic charcoal trap and released into the noble gas mass spectrometer for 50 cycles of $^3He/^4He$ analyses. Experimental errors of $^3He/^4He$ and $^4He/^{20}Ne$ ratios are 0.4% and 3%, respectively, at $1\sigma$ level[30]. The $\delta^{13}C$ values of $CO_2$ and $CH_4$, and $\delta^{15}N$ values of $N_2$ were measured by a continuous flow GC-IRMS system (IsoPrime100 equipped with a vario-EA system). The overall error of analyses is approximately 0.3‰ at $2\sigma$ level[65].

**Data compilation and processing**. The $^3He/^4He$ and $\delta^{13}C–CO_2$ data were compiled from previous studies based on careful evaluation of data quality (Supplementary Data 2). Together with data reported in this study, we calculated the average $^3He/^4He$ value of each sampling site (Supplementary Data 3) to avoid the

influence of repeated sampling of the same site on the statistical tests. Geodetic strain rate data were compiled for the SETP and adjacent region based on Global Strain Rate Model[46]. GPS coordinates were considered for selection of strain rate for individual sampling site. Sample distance to the IACB, the first-order boundary between the Indian and Asian continent as marked by the southern thrust front of the Himalayas and the western thrust front of the Myanmar fold-and-thrust belt (Fig. 1), was measured for each sampling site by Google Earth software. In this case, each sampling site corresponds with one total strain rate, one averaged $^3$He/$^4$He value, and one sample distance to the IACB (Supplementary Data 3), by which average values of strain rate, $^3$He/$^4$He value, and sample distance to the IACB were calculated for the active faults in non-volcanic regions. Pearson correlation tests were used to evaluate potential correlations between corresponding variables as shown in the main text. More details on methods for He isotope correction and calculation, compilation of strain rate data, and ternary mixing models presented in Fig. 3 are provided in Supplementary Information.

## Data availability

The data that support the findings of this study are available in Supplementary Data 1–5. Total strain rates used to generate Fig. 4a are provided as a Source Data file. All the data have been deposited in GitHub repository (https://github.com/mzhangrocks/Plateau-Growth).

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

## Acknowledgements

This work was supported by China Seismic Experimental Site (CSES) (2019CSES0104), the Strategic Priority Research Program (B) of Chinese Academy of Sciences (XDB26000000), the National Key Research and Development Program of China (2020YFA0607700), the National Natural Science Foundation of China (41930642, 41602341, 41772355, and 41702361), the Second Tibetan Plateau Scientific Expedition and Research Program (STEP) (2019QZKK0702), and the United Laboratory of High-Pressure Physics and Earthquake Science (2019HPPES02). P.H.B. was supported by the US National Science Foundation EAR Grant 1144559 during a portion of this work. Field assistance from S. Gao, D. Yang, Y. Yang, and L. Zhao, and laboratory assistance from K. Blackmon, B. Deck, and Z. Gao are appreciated. We dedicate this work to the memory of the late David R. Hilton who wisdom, inspiration, and selflessness benefited so many and is sadly missed.

## Author contributions

M.Z., Z.G., and S.X. designed research; M.Z., L.Z., P.H.B., S.X., Y.S., S.A.H. A.-T.C., Z.C., C.-Q.L., S.-L.L., Y.-C.L., G.Z., Z.L., L.L., and Y.L. performed research; M.Z., Y.S., Z.G., S. X., P.H.B., L.Z., and S.A.H. analyzed data; and M.Z. wrote the paper with contributions from the coauthors.

## Competing interests

The authors declare no competing interests.
