## [Peer Review File · Nature Communications]

REVIEWER COMMENTS

Reviewer #1 (Remarks to the Author):

- What are the noteworthy results?

This work clearly highlights that He-CO₂-N₂ measurements of hydrothermal fluids show a zone of deep-sourced volatiles emissions in conjunction with active faulting and magma degassing in southeastern Tibetan Plateau. This differentiates between the crustal only plateau growth hypothesis and the extrusion of micro-continental blocks along strike-slip faults that may extend into the lithospheric mantle.

The clear and consistent identification of mantle volatiles throughout several fault blocks in the southeastern Tibetan Plateau by the authors appears to confirm the later hypothesis.

- Will the work be of significance to the field and related fields?

I'm not overly familiar with the literature for this region, but it appears to be a robust study comprised of a significant amount of new data and a QC'd compilation of existing measurements.

If the resolution of the different models is as clear cut as the authors suggest and this was not known before in this portion of the Tibetan Plateau, then it has the potential to be significant to workers in this region and others investigating other uplifted orogenic plateaus, particularly those that do not show sharp topographic gradients.

How does it compare to the established literature? If the work is not original, please provide relevant references.

I'm not overly familiar with the literature for this region, but it appears to be a robust study comprised of a significant amount of new data and compilation of existing measurements. The identification of mantle volatiles in regions of significant uplift appears to be common with previous work in other uplift regions such as the Colorado Plateau, and strongly suggests that the sublithospheric mantle plays a role in the uplift of these regions.

- Does the work support the conclusions and claims, or is additional evidence needed?

In the most part, although there is some ambiguity in the term helium degassing rate which is used throughout the study. The authors use this to refer to increased ³He/⁴He ratios. I recommend that this is changed to refer to increased ³He content, or that the authors explain more clearly what they are referring to in terms of rate.

I'm also not an expert in the strain rate calculation but the methods used appear to be robust, and the match of high strain rate to elevated ³He/⁴He ratios in the Kangding-Moxi region

- Are there any flaws in the data analysis, interpretation and conclusions? - Do these prohibit publication or require revision?

Not significant flaws from what I can see – though I strongly recommend that the figure captions are expanded, as they currently do not explain to the reader what they are meant to take from the individual figures, which are all quite complex and difficult to interpret without further guidance. This is particularly the case with Figure 5, which is a good attempt to visually show the model that the authors wish to convey to the reader, but is very difficult to make sense of.

Figure 1 shows the clear areas of elevated and non-elevated $^3\text{He}/^4\text{He}$ ratios, and I am interested to know how the authors account for this observation in their model. This is not a major point, as it is clear that there is abundant mantle influence along several of the faults in the SETP.

Finally, it would be helpful if the authors improved their description of the types of wells sampled in the study, as the current description of 'drilling wells' is not very clear – are these drinking water wells or something else?

- Is the methodology sound? Does the work meet the expected standards in your field?

The methodology is sound and analysis has been undertaken within well-established laboratories, which have strong track-records for such measurements.

- Is there enough detail provided in the methods for the work to be reproduced?

For the most part there is, although it would be tricky to collect this amount of data from what I imagine is a difficult area to access – so the authors should be commended on the extent of the sampling programme and resulting dataset.

Stuart Gilfillan

University of Edinburgh

Reviewer #2 (Remarks to the Author):

– General Comments

The authors collected and analyzed subsurface fluid samples from the southeastern Tibetan Plateau (SETP) for their geochemical characteristics including helium, and stable carbon and nitrogen isotopes. Combined with the compiled previously reported geochemistry data in the same region as well as the geodetic data, the authors have presented original and convincing multi-lines of evidence for 1) the mantle-involved mechanism is the fundamental driving force of the lateral expansion and localized surface uplift in the SETP as well as 2) the westward rollback of the Indian slab leading to the strain reorganization in SETP. In particular, this interdisciplinary work largely contributes to the debate around the mechanism of SETP growth. The reviewer has no major concerns about this manuscript. Instead, the reviewer would like a few minor issues to be clarified or fixed prior to the publication of this work. These minor issues are listed below.

– Detailed Comments

1. Lines 32-34 and Lines 323-325: It is unclear to the reviewer what this sentence of "Such links between.....be dominant" means exactly. Maybe change 'dominant' to 'plausible'? or change to "volatile emission is inferred to be largely driven by plateau growth....."?
2. Lines 86-89: This long sentence does not make a lot of sense logically. Why would the link between volatile emission and plateau growth infer the control of India-Asia convergence on plateau growth? Either of these statements might be true. But do they have causal relation?
3. Line 102: To help readers who are not familiar with noble gas geochemistry, it will be nice for the authors to provide detailed equations and descriptions (in SI or method) wrt how the authors to correct R/Ra for air-component. Neither the 'Method' nor the 'SI' provides such information.
4. Line 122: A similar comment as the one above on 'Rayleigh distillation'. Although the SI has a dedicated section for 'Rayleigh distillation', not sufficient information has been provided to enable the reproduction of the results. For example, is such Rayleigh distillation a boiling model, diffusion-controlled process, or solubility-controlled process? This is unclear. Are the authors modeling the residual phase or the escaped phase? Etc etc. More details are warranted to improve the reproductivity of this work.
5. Lines 124-125, Line 131, Line 150: A simple Pearson or Spearman test should be done to justify/disprove the correlation mentioned here. In particular, the so-called negative correlation in Fig. 2A is not obvious to the reviewer.
6. Lines 151-152: There is a leap from this absence of a hydrothermal degassing system to the lack of correlation. The relationship between these two needs to be fully explained.
7. Lines 166-169: Such a positive correlation between R/Ra ratio and distances to IACB is very interesting if true. A statistical test is warranted to confirm this correlation.
8. Line 196 'biogenic carbon': Please clarify what type of carbon the 'biogenic' is referring to. Why would this be biogenic carbon? Could it be a thermogenic carbon, considering many samples are from the hydrocarbon-rich Sichuan Basin?
9. Line 198 "The available nitrogen isotope data are not enough": do the author refer to the number of samples or the nature of the data?
10. Lines 213-215: 1) a simple Pearson or Spearman test should be done to justify the positive correlation in Figure 4B and 4C; 2) the statement of strain rate increases as moving away from the IACB is not clearly supported by Figure 4. A plot of strain rate versus distance will be helpful along with statistical tests.
11. Line 231 "diffusively distributed convergence force": The reviewer does not understand this phrase. Please clarify.
12. Fig. 4A: the location of EHS should be marked.

Tao Wen

Department of Earth and Environmental Sciences

Syracuse University

Reviewer #3 (Remarks to the Author):

Zhang et al.

Linking deep-sourced 1 volatile emissions to plateau growth dynamics in southeastern Tibetan Plateau

The manuscript deals with the interesting topic of the India-Asia collision and the growth of the Tibetan Plateau, focusing on Southeastern Tibet and the smooth topographic gradient of this part of the plateau with the surrounding lowlands, which is in stark contrast to the steep plateau edges found in the north, south, and east. The authors present He-CO₂-N₂ systematics of hydrothermal fluids to gain insight into the depth of tectonic activity in the region (crustal scale or lithospheric scale).

The most robust conclusions from this work are: (1) that tectonic deformation in the region along major strike slip faults (e.g. Xianshuihe fault) involves the entire lithosphere (crust and lithospheric mantle) and is not limited to the crust, and (2) that there is a positive correlation between crustal strain rate and helium degassing rate.

The authors draw a number of other conclusions and propose a variety of other things, but these are much less constrained, sometimes self-evident, or lack novelty. For example, the authors argue in the abstract that their investigation provides "direct evidence for the control of India-Asia convergence on plateau growth". First of all, by using He-CO₂-N₂ systematics of hydrothermal fluids you cannot provide "direct" evidence, only indirect evidence. Second, it is clear to everyone in the Solid Earth community that plateau growth is controlled and driven by India-Asia convergence. See more comments below.

In addition to the above reservations, the text is in many places difficult to follow and when tectonics and forces are discussed the text is often unclear or ambiguous. The English also needs considerable improvement.

COMMENTS

32-34: "Such links between deep-sourced volatile emissions and plateau growth dynamics are inferred to be dominant since the mid to late Miocene based on regional tectono-magmatic history and plate reconstruction model."

At many places in the text, such as in the above sentence from the abstract, the authors mention how their work has implications for, and provides constraints on, plateau growth, i.e. its lateral growth. However, none of their data, nor their interpretations, provide any actual information or quantitative estimates on this. In which direction is the SE Tibetan Plateau growing? At what rate? How did you deduce this from your He-CO₂-N₂ systematics? None of these questions are addressed nor answered in the manuscript. Also, there is no plate reconstruction model presented in this manuscript.

The first sentence of Fig. 5 reads "Schematic model showing growth of the SETP since the mid to late Miocene." However, no growth is presented/illustrated in this figure. Panels b and c show 3D sketches of the structural setting of the SETP, while panel a shows a schematic cross-section of the Indian (Myanmar) subduction zone. The proposed slab rollback in this diagram is not rollback but merely slab steepening, and the evidence (assumption) for flat slab subduction at 13-10 Ma is not presented (justified).

First paragraph of the introduction: The authors do not make clear at all how deep-sourced volatile emissions can provide any constraints on plate/mantle scale geodynamic processes and might help elucidate the main driving mechanism of plateau building and orogenesis. This makes the manuscript not really accessible to readers who are not familiar with He-CO₂-N₂ systematics of hydrothermal fluids.

50-57: The authors argue that the Southeast Tibetan Plateau is an ideal place to study plateau growth because it has a low topographic gradient spread out over 1000-1500 km, in contrast to the northern, southern and eastern margins with much steeper topographic gradients, but they do not explain why, they provide no rationale. Indeed, one could argue exactly the opposite, namely that lateral expansion of a plateau can be tracked more accurately when the edge of the plateau is clearly marked (with a steep edge) than when the edge is not clearly defined (i.e. with a broad, low-angle edge).

90-92: This sentence is totally unsupported. Also, one would expect surface subsidence, not uplift, with

westward slab rollback due to the extension in the overriding plate.

139-141: But if these are all normal fault and thrust fault settings, then how can it be analogous to the SETP, which is a zone dominated by strike-slip faulting, as shown in Fig. 1?

159-164: The authors define an India-Asia convergence boundary (IACB), plotted in Fig. 1, and use it as a spatial reference against which the different sample groups are plotted in Fig. 3. The location of the IACB, however, is tectonically incorrect. Indeed the boundary should be located at the southern thrust front of the Himalaya and the western thrust front of the Myanmar fold-and-thrust belt. This is where the plate boundary lies, not several 100s of km to the east.

183-188: The comparison between Tengchong volcanoes-IACB distance and trench-arc distance in Japan of ~300 km is baseless and unjustified, because the location of the IACB as proposed by the authors is incorrect. Indeed, the IACB in the Myanmar region should be at the western edge of the Myanmar fold-and-thrust belt, which is located up to 400 km further to the west, making the Tengchong volcanoes-IACB distance some 700 km.

190-191: If this region is characterised by E-W extension, then why are there only strike-slip faults and no normal faults reported on the map in Fig. 1?

213-215: This is at least partly due to the strange location of the IACB. The highest strain rates are actually located at the India-Asia plate boundary, which is located west of 96 degrees East, which is not shown on the map.

230-231: "a more diffusively distributed convergence force" What is this? I've never come across such terminology in the geophysics literature.

235-236: This is more related to the rheology of the crust/lithosphere.

A general comment, the results section is full of interpretations and speculations. These all belong to the discussion section.

256-260: The assumption of a constant stress field since 13-10 Ma is ill-founded., because it does not depend on the convergence rate and direction. It actually depends on the rate and direction of plate boundary migration. As such, the suggestion of a steady helium degassing since the mid-late Miocene is not supported.

290-293: Why is rollback required? What is the role of rollback in driving the magmatism?

320: What is "the distribution of India-Asia convergence"? This is entirely unclear.

325-327: "We thus propose that late Cenozoic growth (ca. 13–10 Ma to the present) of the SETP can be best explained by a strain reorganization in response to westward rollback of the Indian slab."
You have not provided any evidence that the Indian slab has been rolling back westward since 13-10 Ma.

363-368: Then where are the data from the earthquake catalogue and the seismic images to determine the location of the IACB? Show then in maps/figures.

This is in disagreement with your data availability statement, which says: "All data needed to evaluate the conclusions in the paper are present in the paper and/or the Supplementary Information."

373-375: The first order boundary is the plate boundary. And what is a "convergence force"?

590-591: Extrusion and rotation are not driven by strain! They are driven by forces and stresses.

Response to Reviewers' Comments

Notes on the point-by-point responses

Reviewers' Comments: Plain text, Calibri font

Authors Responses: *Italic, indented dark red text, Calibri font*

Reviewer #1

– What are the noteworthy results?

This work clearly highlights that He-CO₂-N₂ measurements of hydrothermal fluids show a zone of deep-sourced volatiles emissions in conjunction with active faulting and magma degassing in southeastern Tibetan Plateau. This differentiates between the crustal only plateau growth hypothesis and the extrusion of micro-continental blocks along strike-slip faults that may extend into the lithospheric mantle.

The clear and consistent identification of mantle volatiles throughout several fault blocks in the southeastern Tibetan Plateau by the authors appears to confirm the later hypothesis.

***Response:** We thank the reviewer for the detailed and constructive comments, and appreciate that the reviewer recognizes the significance of our work. In short, our findings provide geochemical evidence for constraining the depths of volatile sources and associated deep dynamics (i.e., mantle-scale vs. crustal-scale) contributed to growth of the SETP.*

– Will the work be of significance to the field and related fields?

I'm not overly familiar with the literature for this region, but it appears to be a robust study comprised of a significant amount of new data and a QC'd compilation of existing measurements.

If the resolution of the different models is as clear cut as the authors suggest and this was not known before in this portion of the Tibetan Plateau, then it has the potential to be significant to workers in this region and others investigating other uplifted orogenic plateaus, particularly those that do not show sharp topographic gradients.

***Response:** Contrasting end-member models for the growth of the Tibetan Plateau have long been debated, as reviewed by Tapponnier et al. (2001, Science, <https://doi.org/10.1126/science.105978>), Royden et al. (2008, Science, <https://doi.org/10.1126/science.1155371>), and Hubbard & Shaw (2009, Nature, <https://doi.org/10.1038/nature07837>). Significantly, this work provides the key geochemical constraints needed to discriminate between the two end-member models of plateau growth.*

– How does it compare to the established literature? If the work is not original, please provide relevant references.

I'm not overly familiar with the literature for this region, but it appears to be a robust study comprised of a significant amount of new data and compilation of existing measurements. The identification of mantle volatiles in regions of significant uplift appears to be common with previous work in other uplift regions such as the Colorado Plateau, and strongly suggests that the sublithospheric mantle plays a role in the uplift of these regions.

***Response:** We agree and thank the reviewer for their constructive and supportive comments.*

– Does the work support the conclusions and claims, or is additional evidence needed?

In the most part, although there is some ambiguity in the term helium degassing rate which is used throughout the study. The authors use this to refer to increased $^3\text{He}/^4\text{He}$ ratios. I recommend that this is changed to refer to increased ^3He content, or that the authors explain more clearly what they are referring to in terms of rate.

I'm also not an expert in the strain rate calculation but the methods used appear to be robust, and the match of high strain rate to elevated $^3\text{He}/^4\text{He}$ ratios in the Kangding-Moxi region.

***Response:** As suggested by the reviewer, we have removed any mention of 'helium degassing rate' from the revised manuscript. Instead, we simply take the $^3\text{He}/^4\text{He}$ value as a proxy of deeply-sourced He degassing. Then, we calculate the proportions of MORB mantle He and crustal He, using air-corrected $^3\text{He}/^4\text{He}$ to evaluate He origin and the levels of mantle He degassing.*

The compiled strain rate data in this study are from Kreemer et al. (2014, G-cubed, <https://doi.org/10.1002/2014GC005407>), and are consistent with other geodetic studies such as Li et al. (2019, EPSL, <https://doi.org/10.1016/j.epsl.2019.07.010>). Our finding that high strain rates match elevated $^3\text{He}/^4\text{He}$ values in the Kangding-Moxi region is indeed consistent with that observed in Basin and Range Province, western North America (Kennedy & van Soest, 2007, Science, <https://doi.org/10.1126/science.1147537>). This may be a common feature for mantle He degassing from active faults, where the fault permeability is influenced by the stresses exerted on the faulting system.

– Are there any flaws in the data analysis, interpretation and conclusions? - Do these prohibit publication or require revision?

Not significant flaws from what I can see – though I strongly recommend that the figure captions are expanded, as they currently do not explain to the reader what they are meant

to take from the individual figures, which are all quite complex and difficult to interpret without further guidance. This is particularly the case with Figure 5, which is a good attempt to visually show the model that the authors wish to convey to the reader, but is very difficult to make sense of.

Response: *Thanks for suggestion, and we agree with the reviewer. We have revised the figures and expanded the figure captions to make it more understandable to the readers. Necessary explanation is detailed for all the figures, especially Figure 5.*

Figure 1 shows the clear areas of elevated and non-elevated $^3\text{He}/^4\text{He}$ ratios, and I am interested to know how the authors account for this observation in their model. This is not a major point, as it is clear that there is abundant mantle influence along several of the faults in the SETP.

Response: *Using our He data, we identified three regions of elevated $^3\text{He}/^4\text{He}$ in the SETP, which are: (i) Kangding-Moxi section of the XSHF ($^3\text{He}/^4\text{He} = 0.82\text{--}3.79 R_A$ and 10–47% mantle He), (ii) an area within ~60 km from Quaternary volcanoes in the Tengchong block ($^3\text{He}/^4\text{He} = 1.09\text{--}5.92 R_A$ and 13–74% mantle He), and (iii) an area within ~40 km from Quaternary volcanoes in the Simao block ($^3\text{He}/^4\text{He} = 1.04\text{--}5.39 R_A$ and 13–68% mantle He). The other active faults in non-volcanic regions are characterized by relatively low $^3\text{He}/^4\text{He}$ values ($< 1R_A$). The details about the $^3\text{He}/^4\text{He}$ distribution can be found on Lines 122–165 of the revised manuscript.*

The elevated $^3\text{He}/^4\text{He}$ values in Kangding-Moxi section are attributed to high shear strain rates along bend section of the XSHF, which has enhanced fault permeability and allows rapid release of mantle He without significant contamination by crustal He. The elevated $^3\text{He}/^4\text{He}$ values observed in the sampling sites close to Tengchong and Simao Quaternary volcanoes can be explained by degassing of mantle-derived magma bodies beneath the volcanoes. Previous volcanological and geophysical studies have suggested the presence of possible magma chambers beneath Quaternary volcanoes in the Tengchong (Hua et al., 2019, JVGR, <https://doi.org/10.1016/j.jvolgeores.2019.04.002>) and Simao (Cheng et al., 2019, Tectonophysics, <https://doi.org/10.1016/j.tecto.2019.04.032>) block.

In contrast, the non-elevated $^3\text{He}/^4\text{He}$ values are attributable to high degrees of crustal contamination for the mantle He. This includes the following two cases: (i) the fluids discharging >40–60 km away from Quaternary volcanoes have long transport path/time from the source to the surface and experience high degrees of crustal contamination; and (ii) low strain rates and thus weak fault fragmentation and low fault permeability of the active faults would lead to high degrees of crustal contamination in the non-volcanic regions.

Finally, it would be helpful if the authors improved their description of the types of wells sampled in the study, as the current description of ‘drilling wells’ is not very clear – are these drinking water wells or something else?

Response: Thank you for this suggestion. Our sampling sites include both water wells (for earthquake observation) and gas wells (for natural gas exploration). We have improved the descriptions in the Supplementary Information, as per the reviewer's suggestion.

– Is the methodology sound? Does the work meet the expected standards in your field?

The methodology is sound and analysis has been undertaken within well-established laboratories, which have strong track-records for such measurements.

Response: We agree.

– Is there enough detail provided in the methods for the work to be reproduced?

For the most part there is, although it would be tricky to collect this amount of data from what I imagine is a difficult area to access – so the authors should be commended on the extent of the sampling programme and resulting dataset.

Response: Thanks for recognizing the vast amount of work that went into this project. Indeed, these results are the culmination of several extensive sampling campaigns as well as a large amount of data, which is compiled from the literature.

– Detailed Comments

1. Lines 34–36: “...reveal a unique role of deep Earth degassing in gaining insights into the driving force for orogenic plateau growth...”. What is this role and what is the evidence to support this statement - it would be helpful if the authors could be more specific.

Response: The role of deep Earth degassing refers to depths of degassing (i.e., mantle-scale vs. crustal) and how this relates to the underlying geodynamics involved in plateau growth, which can be qualitatively assessed by coupling volatile constraints with geophysical observations. The most convincing evidence is the simple fact that $^3\text{He}/^4\text{He}$ values in hydrothermal fluids are effective in tracing the mantle-derived components.

We have rewritten the abstract section, in which we emphasize the effectiveness of the deeply-sourced volatiles in constraining the deep dynamic processes involved in orogenic plateau growth. Please see Lines 34–39 of the revised manuscript.

2. Line 39: “...such as Altiplano, Colorado, and Tibet...”. Also resolved in CO₂ well gases, e.g., Gilfillan et al. 2017 <https://www.sciencedirect.com/science/article/pii/S0009254117305296> Gilfillan et al., 2008 <https://www.sciencedirect.com/science/article/abs/pii/S0016703707005807> and Ballentine et al., 2005 <https://www.nature.com/articles/nature03182> to name three

Response: Thanks for suggestion. We have added the above references.

3. Lines 43–44: “...would facilitate a better understanding of the plateau growth dynamics over geological timescales...”. In what way - can this be made clearer to the reader?

Response: We have added the explanation for how to constrain plateau growth dynamics using volatile geochemistry on Lines 57–73 of the revised manuscript. For the interpretation of plateau growth over geological timescales, please see Lines 314–366 of the discussion section.

4. Line 45: “...driving force that dominated plateau growth...”. Is this referring to Plateau uplift or extension or both?

Response: In the case of the Tibetan Plateau, the plateau growth processes are referring to both surface uplift and lateral expansion, which are ultimately driven by the India-Asia convergence, including the continental collision and subduction processes.

5. Lines 56–57: “...via lateral expansion and surface uplift...”. This could come earlier.

Response: Thanks for this suggestion. We have reorganized the Introduction section. The styles of orogenic plateau growth are now described in the first sentence of Introduction (Lines 43–44 of the revised manuscript).

6. Lines 85–86: “...allow us to better understand lithospheric structure of the SETP, and to further constrain the depths of plateau growth dynamics...”. This could be more specific.

Response: Thanks for suggestion. We have rewritten the Introduction section to make it more specific and understandable. This sentence is not used in the revised Introduction.

We suggest that evidence from volatile geochemistry (especially $^3\text{He}/^4\text{He}$) is capable of revealing the depths of fault penetration (i.e., lithospheric-scale vs. crustal-only scale). Considering that the plateau growth processes are accomplished by movements along the faults, there are potential links between volatile geochemistry and plateau growth dynamics. This is the motivation of our study.

7. Line 88: “... $^3\text{He}/^4\text{He}$ distribution (i.e., helium degassing rate)...”. This is not really a degassing rate - it is a ratio of the crustal and mantle derived helium isotopes.

Response: Thanks for pointing out this unclear definition. In the revised manuscript, we use air-corrected $^3\text{He}/^4\text{He}$ values to calculate the mantle contributions (expressed in % mantle He). The calculated results are used to reflect whether or not there are clear mantle inputs into the deeply-sourced fluids. In this case, we have removed any mention of ‘helium degassing rate’ from the revised manuscript.

8. Line 96: Change “...drilling wells...” to “...drilled wells...”. Water wells or gas wells or both?

Response: *We have changed “drilling” to “drilled” in corresponding places of the revised manuscript (i.e., Lines 94 and 382). Our sampling sites include both water wells (for earthquake observation) and gas wells (for natural gas exploration). Detailed description of the drilled wells is provided in the Supplementary Information.*

9. Lines 129–130: “...involvement of organic contamination and elemental/isotopic fractionation...”. Has the role of solubility fractionation during phase separation been examined? Recent work by Karolyte et al., found that this is responsible for low ^3He concentrations and associated high $\text{CO}_2/{}^3\text{He}$ in CO_2 springs in SE Australia, which are commonly interpreted as evidence for mixing with crustal CO_2 .

<https://www.sciencedirect.com/science/article/pii/S001670371930331X>

Response: *Indeed, there is a negative correlation between $\text{CO}_2/{}^3\text{He}$ and He contents for the hydrothermal fluid samples. Considering the length of main text, the discussion on the role of solubility fractionation during phase separation is provided in Supplementary Information (Supplementary Fig. 6). We have added Karolyte et al. (2019) to refs. list.*

10. Line 141: “...and the East African rift...”. Also recently documented along a fault in South Africa - <https://www.nature.com/articles/s41467-019-12944-6>

Response: *Thanks for suggestion. We have added this reference to illustrate the release of deeply-sourced volatiles (e.g., He and CO_2) from active faults in Line 59 of the revised manuscript.*

11. Lines 156–157: “...significant discharge of mantle-derived fluids in an area of ~1,200 km by ~300–600 km (Fig. 1)...”. I would disagree with significant in this statement - there is one fault and one block which shows ${}^3\text{He}/{}^4\text{He}$ ratios above 1 - the rest do not have a significant mantle component. The two hotspots are the TVF and the Xianshuihe fault. There are above crustal values in the Bookshelf, Litang and three rivers faults, but it would be more accurate to describe these as elevated relative to crustal ratios rather than significant.

Response: *Thanks for comment and suggestion. Originally, we followed Crossey et al. (2009) to describe the extent of mantle degassing in the SETP as significant (Fig. R1), because about >70% of the sampling sites have ${}^3\text{He}/{}^4\text{He}$ values $>0.2 R_A$.*

We agree with the reviewer that it is more appropriate to use significant to describe the ${}^3\text{He}/{}^4\text{He}$ hotspots in the SETP. In the revised manuscript (Lines 122–129), we identified three hotspots using new dataset of ${}^3\text{He}/{}^4\text{He}$ values, which are bend section of the Xianshuihe fault, Tengchong volcanoes, and Simao volcanoes.

To make the description of He isotope data more accurate, we calculated the relative proportions of crust and mantle He. Detailed information about the He isotopes is given

in "Origin of helium" of the Result section.

EXPLANATION	
Helium isotope analysis (R_A)	
○	< 0.09 No significant mantle component
○	0.1 - 0.9 Significant mantle component
○	1.0 - 1.9 High mantle component
○	2.0 - 3.9 Very high mantle component
○	4.0 - 8.0 Near-MORB mantle component
●	Travertine-depositing spring

Fig. R1. Explanation for helium isotope analysis quoted from Crossey et al. (2009) <https://pubs.geoscienceworld.org/gsa/gsabulletin/article-abstract/121/7-8/1034/2380>

12. Line 189: "...helium degassing rates...". Relative mantle contribution, not rates.

Response: Yes, it is relative mantle contribution. we have removed any mention of 'helium degassing rate' from the revised manuscript.

13. Line 198: "...There is insufficient resolution in the available nitrogen isotope data to...".

Response: Thanks for suggestion. We have revised the Result section. This sentence about the paucity of data will not be used in the revised manuscript.

14. Lines 201–203: "...the air-corrected $^3\text{He}/^4\text{He}$ value is more effective in characterizing the spatial variations in deep-sourced volatile emissions...". This is not surprising!

Response: Yes, the $^3\text{He}/^4\text{He}$ evidence is robust for tracing mantle He degassing.

15. Line 209: "...Combined with helium degassing rates...". helium $^3\text{He}/^4\text{He}$ ratios.

Response: Yes, not helium degassing rates. We have simply used $^3\text{He}/^4\text{He}$ ratios, and removed "helium degassing rate(s)" from the revised manuscript

16. Lines 248–250: "...Our $^3\text{He}/^4\text{He}$ data would thus provide direct evidence for the involvement of mantle dynamics in southeastward growth of the SETP...". This certainly appears to be the case.

Response: Thanks for acknowledging this. Mantle dynamics are important for the growth of the SETP in terms of lateral expansion and localized surface uplift.

17. Lines 266: "...volatile emissions over geological timescales...". I don't quite follow this -

can more detail be given, as i think that there is a link missing somewhere.

Response: *We have rewritten the Discussion section to make it more understandable. As for volatile emissions over geological timescales, please see Lines 314–366 of the revised manuscript.*

18. Lines 273–274: “..., the initiation ages of these active faults are confined to the mid to late Miocene time...”. How are these ages obtained?

Response: *The initiation ages of active faults are obtained from geological observations and several dating methods, including low-temperature thermochronology [e.g., fission-track, (U-Th)/He], radiometric dating of whole-rock and minerals (e.g., U-Pb, K-Ar, ⁴⁰Ar-³⁹Ar), magnetostratigraphy, and structural analysis. This is provided in Supplementary Table 2.*

19. Line 299: “...the significant proportion of mantle volatile that are degassing...”.

Response: *Thanks for suggestion. We have revised this sentence. Please see Lines 294–296 of the revised manuscript.*

20. Line 308: “...high helium degassing rates...”. Again please reword.

Response: *We have deleted the “helium degassing rates” in the revised manuscript.*

21. Line 334: “...drilling wells...”. As per previous comments - please provide more detail on the wells sampled.

Response: *Thanks for suggestion. Detailed description of the drilled wells is provided in the Supplementary Information.*

22. Line 363–365: “...We used the earthquake catalogue...”, “...The earthquake catalogue of the Myanmar...”.

Response: *We have revised the location of the IACB according to the suggestion of Reviewer #3. Therefore, the earthquake catalogue is not used in the revised manuscript.*

23. Line 539: Fig. 1 - It would be helpful to the reader if the acronyms for the faults shown on the next figures could be added to this figure.

Response: *Thanks for suggestion. We have added the acronyms of sample groups to Fig. 1, including active faults, Quaternary volcanoes, and the Sichuan basin.*

24. Line 585: Fig. 5c - I'm struggling to understand this diagram - is it a cross-section? If where is it focused? Same question applies if it is a plan view.

Response: We are sorry for the unclear information presented in this figure. We have revised the figure (e.g., removing Panel A and Keeping Panels B & C) and expanded the caption. In the revised version of Fig. 5, the original Panel C is changed to Panel B (i.e., Fig. 5B).

The location of Fig. 5B is marked in Fig. 5A. It is used to display an overlooking view of the Kangding-Moxi section of the XSHF. The surface and Moho interface are shown to illustrate a 3D structure of this diagram. The fault plane of the XSHF extends to lithospheric mantle. The bend section has experienced transpressional uplifting due to the confluence of the XSHF and the ANHF (Anninghe fault), which created the Gongga Shan massif and favorable conditions for rapid release of mantle He.

Reviewer #2

– General Comments

The authors collected and analyzed subsurface fluid samples from the southeastern Tibetan Plateau (SETP) for their geochemical characteristics including helium, and stable carbon and nitrogen isotopes. Combined with the compiled previously reported geochemistry data in the same region as well as the geodetic data, the authors have presented original and convincing multi-lines of evidence for 1) the mantle-involved mechanism is the fundamental driving force of the lateral expansion and localized surface uplift in the SETP as well as 2) the westward rollback of the Indian slab leading to the strain reorganization in SETP. In particular, this interdisciplinary work largely contributes to the debate around the mechanism of SETP growth. The reviewer has no major concerns about this manuscript. Instead, the reviewer would like a few minor issues to be clarified or fixed prior to the publication of this work. These minor issues are listed below.

Response: We would like to thank the reviewer for their detailed and constructive comments, and we appreciate that the reviewer recognizes the significance of our work and gives their endorsement.

– Detailed Comments

1. Lines 32–34 and Lines 323–325: It is unclear to the reviewer what this sentence of “Such links between.....be dominant” means exactly. Maybe change ‘dominant’ to ‘plausible’? or change to “volatile emission is inferred to be largely driven by plateau growth.....”?

Response: We have rewritten the abstract and discussion section, including this sentence, to make it clearer and concise.

Yes, we agree with the reviewer that volatile emission is inferred to be largely driven by plateau growth. The term “links” means that the mantle degassing features in the SETP

are indicative of mantle dynamics involved in plateau growth. We speculate that the mantle degassing features observed at present may have been formed since the beginning of plateau growth in the mid to late Miocene. Our rationale is explained on Lines 314–352 of the revised manuscript.

2. Lines 86–89: This long sentence does not make a lot of sense logically. Why would the link between volatile emission and plateau growth infer the control of India-Asia convergence on plateau growth? Either of these statements might be true. But do they have causal relation?

Response: *Thanks for this comment, we agree it wasn't particularly clear. Our rationale is that if volatile emissions are the direct result of tectonic and magmatic processes involved in plateau growth, then volatile geochemistry data can be used to discuss tectonic and magmatic processes, and constrain the plateau growth dynamics. A more detailed explanation can be found on Lines 57–73 of the revised manuscript.*

The control of India-Asia convergence on plateau growth can be further explained by the revised Fig. 4. We observed positive correlations between $^3\text{He}/^4\text{He}$, strain rate and distance to the IACB along the direction of Indian indentation into Asia (i.e., the northeastward India-Asia convergence). High $^3\text{He}/^4\text{He}$ values are found in the region with high strain rates, suggesting strong stress driven by the India-Asia convergence. In contrast, low $^3\text{He}/^4\text{He}$ values occur in the region with low strain rates and suggest relatively weaker stresses. This difference reveals the variations in distribution of the regional stresses, which have modulated the growth of the SETP. Therefore, this is the link between volatile emissions, plateau growth, and India-Asia convergence. In short, the $^3\text{He}/^4\text{He}$ anomalies are the results of plateau growth driven by India-Asia convergence.

We have rewritten the abstract section to make it clearer to the readers.

3. Line 102: To help readers who are not familiar with noble gas geochemistry, it will be nice for the authors to provide detailed equations and descriptions (in SI or method) wrt how the authors to correct R/Ra for air-component. Neither the 'Method' nor the 'SI' provides such information.

Response: *Thanks for this excellent suggestion. We have added detailed equations and descriptions of He isotope correction and other calculation methods in Supplementary Information.*

4. Line 122: A similar comment as the one above on 'Rayleigh distillation'. Although the SI has a dedicated section for 'Rayleigh distillation', not sufficient information has been provided to enable the reproduction of the results. For example, is such Rayleigh distillation a boiling model, diffusion-controlled process, or solubility-controlled process? This is unclear. Are the authors modeling the residual phase or the escaped phase? Etc etc. More details are warranted to improve the reproductivity of this work.

Response: Thanks for this suggestion. The Rayleigh distillation in the original manuscript is a solubility-controlled process, which is a common process of magma degassing and have been widely studied for active volcanoes. In the original manuscript, we modelled the residual phase in the case of magma degassing. As presented in the response to the next Comment, we found that there is no negative correlation between $^3\text{He}/^4\text{He}$ and $\delta^{13}\text{C}\text{-CO}_2$ values of the Tengchong samples; in this case, the Rayleigh distillation model used in the original version of Fig. 2A would be incorrect or unnecessary to explain the He-C isotope data. Therefore, we will not discuss Rayleigh distillation model and magma degassing of Tengchong volcanoes in the revised manuscript. Accordingly, we decide not to discuss the Rayleigh distillation model in the Supplementary Information, too.

Large variations in the He-CO₂ elemental and isotopic compositions are likely the results of (i) mixing between multiple components (e.g., mantle, carbonate, organic sediments), and (ii) secondary processes that occur in shallow hydrothermal systems. For the latter, the fractionation processes are also controlled by solubility differences in aqueous fluids between He and CO₂. More details on the He-CO₂ elemental and isotopic fractionation are given in the Supplementary Information.

5. Lines 124–125, Line 131, Line 150: A simple Pearson or Spearman test should be done to justify/disprove the correlation mentioned here. In particular, the so-called negative correlation in Fig. 2A is not obvious to the reviewer.

Response: Thanks for suggestion. We have done Pearson test to evaluate the He-C isotope data presented in Fig. 2A of the original manuscript. Indeed, the r value is 0.044, and there is no correlation between $^3\text{He}/^4\text{He}$ and $\delta^{13}\text{C}\text{-CO}_2$ as the reviewer suggested. Thanks for pointing out this error.

We have deleted this section from the revised manuscript. For example, the original Fig. 2A is not used in the revised manuscript. This is because it is not so relevant with our purpose of identifying the release of mantle volatiles from the SETP, especially the active faults along which the Tibetan Plateau expanded (i.e., lateral expansion).

6. Lines 151–152: There is a leap from this absence of a hydrothermal degassing system to the lack of correlation. The relationship between these two needs to be fully explained.

Response: Thanks for pointing this out, as it is indeed quite important. In the revised manuscript, we use the volatile geochemistry data to identify the release of mantle-derived volatiles from active fault zones and Quaternary volcanic fields. We have moved the crustal contamination part based on He-CO₂ systematics to the Supplementary Information.

As presented in the last response to Comment 5, there is no correlation between $^3\text{He}/^4\text{He}$ and $\delta^{13}\text{C}\text{-CO}_2$ of the Tengchong samples. Therefore, we have removed this section from

the revised manuscript. The original version of Figs. 2A, 2B, and 2C are not used in the revised manuscript.

7. Lines 166–169: Such a positive correlation between R/Ra ratio and distances to IACB is very interesting if true. A statistical test is warranted to confirm this correlation.

Response: *Thanks for this suggestion. We have carried out Pearson test for $\log(^3\text{He}/^4\text{He})$ and distance to the IACB. The result show that the r value is 0.737 ($n = 53$) for the sampling sites of the TRF, LTF, and XSHF bend. Please see Lines 139–149 of the revised manuscript.*

8. Line 196 ‘biogenic carbon’: Please clarify what type of carbon the ‘biogenic’ is referring to. Why would this be biogenic carbon? Could it be a thermogenic carbon, considering many samples are from the hydrocarbon-rich Sichuan Basin?

Response: *Apologies, this was a mistake. As shown by the compiled data and also in the revised Fig. 2C, the $\delta^{13}\text{C}\text{-CO}_2$ values of the Sichuan basin samples have a range of -24.2‰ to -11.2‰ , suggesting an organic origin for the CO_2 . We should clarify that the “biogenic carbon” was erroneously used here to explain the $\delta^{13}\text{C}$ values of CO_2 . The organic CO_2 is likely produced by thermogenic decomposition of organic matter. Please see details on Lines 172–176 of the revised manuscript.*

Methane is the most abundant gas species in the Sichuan basin samples, and the $\delta^{13}\text{C}$ values of CH_4 range from -40.3‰ to -31.0‰ . We agree with the reviewer that the CH_4 is thermogenic carbon related with degradation of organic matter in the sedimentary basin setting. Because most samples in the SETP do not have high CH_4 contents, we will not discuss origin of CH_4 in the revised manuscript.

9. Line 198 “The available nitrogen isotope data are not enough”: do the author refer to the number of samples or the nature of the data?

Response: *Thanks for pointing out this unclear sentence. It actually refers to the number of samples that have been analyzed for nitrogen isotopic compositions. We have revised the Result section. This sentence about the paucity of data will not be used in the revised manuscript. In fact, with respect to the hydrothermal fluids in the SETP, our study reports the first set ($n = 34$) of nitrogen isotope data.*

10. Lines 213–215: 1) a simple Pearson or Spearman test should be done to justify the positive correlation in Figure 4B and 4C; 2) the statement of strain rate increases as moving away from the IACB is not clearly supported by Figure 4. A plot of strain rate versus distance will be helpful along with statistical tests.

Response: *Thanks for suggestion. We have used Pearson test to evaluate the correlation.*

For Fig. 4B, the Pearson correlation coefficient is 0.752 (n = 53, where n is the number of sampling sites) for $^3\text{He}/^4\text{He}$ and total strain rate.

We have added the plot of strain rate versus distance as the new Fig. 4C. The Pearson correlation coefficient is 0.810 (n = 53) for distance to the IACB and total strain rate.

The original Fig. 4C is not used in the revised manuscript, because the correlation is not very good. For the sampling sites in all active fault zones of the SETP, we can observe a moderate correlation ($r = 0.483$, $n = 129$) between $^3\text{He}/^4\text{He}$ and strain rates.

11. Line 231 “diffusively distributed convergence force”: The reviewer does not understand this phrase. Please clarify.

Response: *Thanks for pointing out this unclear phrase. Accordingly, we have corrected it in the revised manuscript.*

*The “diffusively distributed convergence force” is an unclear description of the stresses that are distributed within an area, rather than focused in the area. The stresses are provided by India-Asia convergence and are not uniformly distributed in the SETP. This is reflected by the results of several geodetic studies, such as Kreemer et al. (2014, *G-cubed*, <https://doi.org/10.1002/2014GC005407>), Wang & Shen (2020, *JGR Solid Earth*, <https://doi.org/10.1029/2019jb018774>), and Li et al. (2019, *EPSL*, <https://doi.org/10.1016/j.epsl.2019.07.010>). In the case of SETP, the plateau boundary fault is the focal zone of stresses and the total strain rates are high. The plateau interior faults are controlled by relatively distributed stresses and lower total strain rates.*

12. Fig. 4A: the location of EHS should be marked.

Response: *Thanks for suggestion. We have added the location of EHS in Fig. 4A.*

Reviewer #3

Zhang et al. Linking deep-sourced volatile emissions to plateau growth dynamics in southeastern Tibetan Plateau

The manuscript deals with the interesting topic of the India-Asia collision and the growth of the Tibetan Plateau, focusing on Southeastern Tibet and the smooth topographic gradient of this part of the plateau with the surrounding lowlands, which is in stark contrast to the steep plateau edges found in the north, south, and east. The authors present He-CO₂-N₂ systematics of hydrothermal fluids to gain insight into the depth of tectonic activity in the region (crustal scale or lithospheric scale).

The most robust conclusions from this work are: (1) that tectonic deformation in the region along major strike slip faults (e.g. Xianshuihe fault) involves the entire lithosphere (crust and lithospheric mantle) and is not limited to the crust, and (2) that there is a positive correlation between crustal strain rate and helium degassing rate.

***Response:** We would like to take this opportunity to thank the reviewer for detailed and constructive comments, and appreciate that the reviewer recognizes the significance of our work.*

The authors draw a number of other conclusions and propose a variety of other things, but these are much less constrained, sometimes self-evident, or lack novelty. For example, the authors argue in the abstract that their investigation provides "direct evidence for the control of India-Asia convergence on plateau growth". First of all, by using He-CO₂-N₂ systematics of hydrothermal fluids you cannot provide "direct" evidence, only indirect evidence. Second, it is clear to everyone in the Solid Earth community that plateau growth is controlled and driven by India-Asia convergence. See more comments below.

***Response:** Thank you for this constructive criticism, which is well taken. We agree that the term indirect evidence is much more accurate. We have rewritten large sections of the manuscript to make our interpretation and conclusions clearer.*

Again, we totally agree with the reviewer that the He-CO₂-N₂ systematics of hydrothermal fluids can only provide indirect evidence for the plateau growth dynamics. Nevertheless, we would like to emphasize that the evidence from He-CO₂-N₂ systematics (especially ³He/⁴He) is important for constraining the depths of tectonic activities and thus plateau growth dynamics. Therefore, we have revised the manuscript and mainly focused on: (i) the role of He-CO₂-N₂ systematics in identifying the release of mantle volatiles from active faults (i.e., the first two parts of the Results section), and (ii) the spatial correlation between ³He/⁴He and strain rates along the direction of Indian indentation into Asia (i.e., the third part of the Results section). Using these observations, we established the potential links between deeply-sourced He degassing and regional stress field driven by the India-Asia convergence. In the final part, we discussed the possibility of volatile emission responses to initiation of the current stage of plateau growth in the mid to late Miocene.

We agree with the reviewer about the consensus that the growth of the Tibetan Plateau is controlled and driven by India-Asia convergence. Our results would be important for understanding how the India-Asia convergence drives the on-going plateau growth in the SETP and the outgassing of deeply-sourced volatiles. This is briefly explained below.

First, we provide robust evidence for the depths of plateau growth dynamics (i.e., crustal scale vs lithospheric scale). This means that the fluids at the surface are effective in constraining the deep dynamic processes, which can be used to evaluate the different end-member models for plateau growth, such as the lower crustal flow model and

continental extrusion model. It is plausible that both the strike-slip motion (i.e., lateral expansion) and transpressional uplift in the Kangding-Moxi region (i.e., localized surface uplift) are controlled by lithospheric-scale dynamic processes.

Second, as shown by the revised version of Figs. 4B & 4C, the deeply-sourced He degassing patterns agree well with the regional stress field, suggesting the style of stress distribution in the SETP. To be specific, the plateau boundary fault is a focal zone of stresses, while the stresses exerted on the plateau interior are much more distributed. Such stress distribution in the SETP is recorded by our He isotope data. This may reflect how the stresses driven by the India-Asia convergence modulate lateral expansion and localized surface uplift.

In addition to the above reservations, the text is in many places difficult to follow and when tectonics and forces are discussed the text is often unclear or ambiguous. The English also needs considerable improvement.

Response: *We have thoroughly revised the text and English of the manuscript. We hope it is now clearer and more accessible/readable.*

– Detailed Comments

1. Lines 32–34: "Such links between deep-sourced volatile emissions and plateau growth dynamics are inferred to be dominant since the mid to late Miocene based on regional tectono-magmatic history and plate reconstruction model."

At many places in the text, such as in the above sentence from the abstract, the authors mention how their work has implications for, and provides constraints on, plateau growth, i.e. its lateral growth. However, none of their data, nor their interpretations, provide any actual information or quantitative estimates on this. In which direction is the SE Tibetan Plateau growing? At what rate? How did you deduce this from your He-CO₂-N₂ systematics? None of these questions are addressed nor answered in the manuscript. Also, there is no plate reconstruction model presented in this manuscript.

Response: *Thanks for this constructive criticism. We have revised the manuscript, and re-evaluated the implications of our geochemical data for plateau growth.*

We suggest that the role of volatile geochemistry in understanding plateau growth is shown mainly by the following two points: First, the identification of mantle volatile emissions confirms that the depths of plateau growth dynamics are lithospheric-scale. Second, the correlation between ³He/⁴He and strain rates along the direction of Indian indentation into Asia reflects stress distribution in the SETP. These two observations are indicative for understanding how the India-Asia convergence dominates plateau growth in the SETP.

We should clarify that the direction and rate of plateau growth in the SETP cannot be

constrained by He-CO₂-N₂ systematics of the hydrothermal fluids alone, which requires a multidisciplinary approach (geodynamical, geophysical and geochemical observations). With this respect, we suggest that the plate reconstruction model is beyond the scope of our work.

The first sentence of Fig. 5 reads "Schematic model showing growth of the SETP since the mid to late Miocene." However, no growth is presented/illustrated in this figure. Panels b and c show 3D sketches of the structural setting of the SETP, while panel a shows a schematic cross-section of the Indian (Myanmar) subduction zone. The proposed slab rollback in this diagram is not rollback but merely slab steepening, and the evidence (assumption) for flat slab subduction at 13-10 Ma is not presented (justified).

Response: *Thanks for the constructive criticism. We have revised Fig. 5 and the caption to correct those errors.*

First, we have deleted the panel A of Fig. 5. We prefer not to show how the Indian slab experienced rollback in Fig. 5, because we cannot be 100% sure that there was a slab rollback in mid to late Miocene time. Nevertheless, the stress reorganization is certainly required to explain the onset of the on-going plateau growth stage in the SETP. But there may be several causes for this stress reorganization, including but not limited to the slab rollback and trench retreat model. Some previous studies have proposed the rollback and trench retreat model to explain the geological evolution of the SETP, such as Burchfiel & Chen (2013, GSAM, <https://doi.org/10.1130/MEM210>) and Sternai et al. (2014, EPSL, <http://dx.doi.org/10.1016/j.epsl.2014.08.023>). So, in the final part of the revised manuscript, we just cited the above references to show the possibility of Indian slab rollback and trench retreat, rather than showing these processes in Fig. 5. We suggest that this does not change the main conclusions of our work.

The sentence "Schematic model showing growth of the SETP since the mid to late Miocene" is not accurate to be used as the title of Fig. 5. We have changed the title to "Schematic model showing 3D sketches of the structural setting of the SETP and the styles of outward and (localized) upward growth of the Tibetan Plateau". Detailed illustration is provided in the caption of Fig. 5.

First paragraph of the introduction: The authors do not make clear at all how deep-sourced volatile emissions can provide any constraints on plate/mantle scale geodynamic processes and might help elucidate the main driving mechanism of plateau building and orogenesis. This makes the manuscript not really accessible to readers who are not familiar with He-CO₂-N₂ systematics of hydrothermal fluids.

Response: *We have rewritten the Introduction section. Necessary information on how to correlate deeply-sourced volatile emissions with plateau growth dynamics is now presented in the first two paragraphs (Lines 43–73) of the Introduction.*

2. Lines 50–57: The authors argue that the Southeast Tibetan Plateau is an ideal place to study plateau growth because it has a low topographic gradient spread out over 1000-1500 km, in contrast to the northern, southern and eastern margins with much steeper topographic gradients, but they do not explain why, they provide no rationale. Indeed, one could argue exactly the opposite, namely that lateral expansion of a plateau can be tracked more accurately when the edge of the plateau is clearly marked (with a steep edge) than when the edge is not clearly defined (i.e. with a broad, low-angle edge).

Response: *We have added the reason for choosing the SETP as the study area. Please see the third paragraph (Lines 74–89) of the revised Introduction.*

We agree with the reviewer that one could argue that a steep edge of the orogenic plateau is also an ideal place for studying the lateral expansion processes. However, in this study, we focus on the SETP where contrasting plateau growth models (i.e., the lower crustal flow model and continental extrusion model) are debated. This is an important reason for undertaking our work in the southeastern Tibetan Plateau.

3. Lines 90–92: This sentence is totally unsupported. Also, one would expect surface subsidence, not uplift, with westward slab rollback due to the extension in the overriding plate.

Response: *We agree and have deleted the unsupported sentence. In this study, we now only discuss localized surface uplift in the bend section of the Xianshuihe fault. It is difficult to discuss the surface uplift of the other regions based on our data. As the reviewer suggested, we would also not expect surface uplift in the overriding plate in the model of slab rollback.*

4. Lines 139–141: But if these are all normal fault and thrust fault settings, then how can it be analogous to the SETP, which is a zone dominated by strike-slip faulting, as shown in Fig. 1?

Response: *We have deleted this paragraph from the main text. We agree with the reviewer that the types of active faults are different for the Himalayas, central Italy, and the East African rift. The similarity between them was based on the type of volatile degassing. In some regions of the mentioned fault settings, previous studies have found degassing of volatiles that is not related with active volcanoes. This is similar to the non-volcanic regions of the southeastern Tibetan Plateau.*

5. Lines 159–164: The authors define an India-Asia convergence boundary (IACB), plotted in Fig. 1, and use it as a spatial reference against which the different sample groups are plotted in Fig. 3. The location of the IACB, however, is tectonically incorrect. Indeed the boundary should be located at the southern thrust front of the Himalaya and the western thrust front of the Myanmar fold-and-thrust belt. This is where the plate boundary lies, not several 100s of km to the east.

Response: Thanks for pointing out our incorrect interpretation on location of the IACB. We agree with the reviewer and have revised the Figures and related parts in the main text of the revised manuscript. Please see revised version of Figs. 1 and 2.

6. Lines 183–188: The comparison between Tengchong volcanoes-IACB distance and trench-arc distance in Japan of ~300 km is baseless and unjustified, because the location of the IACB as proposed by the authors is incorrect. Indeed, the IACB in the Myanmar region should be at the western edge of the Myanmar fold-and-thrust belt, which is located up to 400 km further to the west, making the Tengchong volcanoes-IACB distance some 700 km.

Response: We agree with the reviewer. We have revised the distance of Tengchong volcanoes to the IACB. Please see revised version of Figs. 2B and 2D. We have deleted the comparison between Tengchong-IACB distance and trench-arc distance in Japan, because it is not so relevant with the theme of the revised manuscript.

7. Lines 190–191: If this region is characterised by E-W extension, then why are there only strike-slip faults and no normal faults reported on the map in Fig. 1?

Response: We have added normal faults in the revised version of Fig. 1. Please see details in the caption of Fig. 1.

8. Lines 213–215: This is at least partly due to the strange location of the IACB. The highest strain rates are actually located at the India-Asia plate boundary, which is located west of 96 degrees East, which is not shown on the map.

Response: We agree with the reviewer that the highest strain rates are located at the fold-and-thrust belts south of the Himalayas and west of the Indo-Burman ranges, which mainly reflect compressional strain in the contact zone between two continental plates. We did not investigate in the strain rates of the India-Asia plate boundary, because it is beyond the scope of our study area. Instead, we mainly focus on the strain rates of the strike-slip fault system in southeastern part of the Tibetan Plateau. We have revised the distances of sampling sites to the IACB. The correlation between distance to the IACB and strain rate still exist ($r = 0.810$). Please see the revised Fig. 4 and Lines 259–264 of revised manuscript.

9. Lines 230–231: "a more diffusively distributed convergence force" What is this? I've never come across such terminology in the geophysics literature.

Response: Thanks for pointing out the unclear use of this terminology. We suggest that the stress exerted on the interior of the SETP is more distributed than that of the boundary. This is consistent with the results of geodetic measurement, which show high strain rates along the plateau boundary and low strain rates within the plateau interior (Kreemer et al., 2014, <https://doi.org/10.1002/2014GC005407>; Li et al., 2019,

<https://doi.org/10.1016/j.epsl.2019.07.010>). We have revised the sentence in several corresponding places (Lines 249, 313, and 348) of the revised manuscript.

10. Lines 235–236: This is more related to the rheology of the crust/lithosphere.

Response: Yes, we agree with the reviewer that India-Asia convergence is dynamically more related to the rheology of the crust/lithosphere. The surface strain rates calculated from geodetic data are likely the manifestations of deformation within limited crustal depths, not the whole crust and/or lithosphere.

It is difficult to correlate our geochemical data to the rheology of the crust/lithosphere. Instead, we found that the $^3\text{He}/^4\text{He}$ values of fluid samples are closely related to total strain rates of active faults, which can be attributed to the influence of fault permeability (usually high strain rates result in enhanced fault permeability) on mantle He degassing. Such phenomenon has also been observed in the Basin and Range Province, western North America (Kennedy & van Soest, 2007, <https://doi.org/10.1126/science.1147537>).

11. A general comment, the results section is full of interpretations and speculations. These all belong to the discussion section.

Response: Thanks for pointing out this problem. We have re-written the results section accordingly. Please see details in the revised manuscript.

12. Lines 256–260: The assumption of a constant stress field since 13-10 Ma is ill-founded., because it does not depend on the convergence rate and direction. It actually depends on the rate and direction of plate boundary migration. As such, the suggestion of a steady helium degassing since the mid-late Miocene is not supported.

Response: We agree with the reviewer and have deleted such assumption and unsupported sentence from the revised manuscript.

13. Lines 290–293: Why is rollback required? What is the role of rollback in driving the magmatism?

Response: The possibility of Indian slab rollback is cited from Burchfiel & Chen (2013, GSAM, <https://doi.org/10.1130/MEM210>). Slab rollback and trench retreat are likely to result in lithospheric extension of the overriding plate, and upwelling + decompressional partial melting of asthenospheric mantle materials, which would cause the magmatism.

14. Line 320: What is "the distribution of India-Asia convergence"? This is entirely unclear.

Response: It should be "distribution of regional stresses". We have revised this in several corresponding places (Lines 249, 313, and 348) of the manuscript.

15. Lines 325–327: "We thus propose that late Cenozoic growth (ca. 13–10 Ma to the present) of the SETP can be best explained by a strain reorganization in response to westward rollback of the Indian slab." You have not provided any evidence that the Indian slab has been rolling back westward since 13-10 Ma.

Response: We have revised the Discussion section, and no more to emphasize the slab rollback and trench retreat model. As a possible explanation for geological evolution in the SETP and adjacent region, the model of slab rollback and trench retreat is cited from previous study of Burchfiel & Chen (2013, GSAM, <https://doi.org/10.1130/MEM210>), which summarized several lines of geological evidence for slab rollback and trench retreat.

16. Lines 363–368: Then were are the data from the earthquake catalogue and the seismic images to determine the location of the IACB? Show then in maps/figures. This is in disagreement with your data availability statement, which says: "All data needed to evaluate the conclusions in the paper are present in the paper and/or the Supplementary Information."

Response: As the location of the IACB has been corrected to the boundary thrust fault between Indian and Asian continent, the data from the earthquake catalogue and the seismic images are not used in the revised manuscript. Therefore, we have deleted these citations in the Methods.

17. Lines 373–375: The first order boundary is the plate boundary. And what is a "convergence force"?

Response: We agree with the reviewer that the first order boundary is the plate boundary (i.e., IACB). We have corrected the location of the IACB according to the reviewer's comment.

The "convergence force" means the driving force provided by India-Asia convergence. We realized that this phrase is not accurate to describe the force exerted in the SETP. Therefore, we have revised the manuscript and simply use "stresses" to explain the driving force for plateau growth.

18. Lines 590–591: Extrusion and rotation are not driven by strain! They are driven by forces and stresses.

Response: Yes, we agree with the reviewer. We have revised the manuscript according to this comment. In the revised manuscript, we used "stresses" to refer to the driving force for extrusion and rotation of the SETP.

REVIEWERS' COMMENTS

Reviewer #1 (Remarks to the Author):

I commend the authors for comprehensively addressing all of my concerns and I am now happy for the paper to be published in Nature Communications without further revision.

Reviewer #2 (Remarks to the Author):

The authors combined the use of helium, and stable carbon and nitrogen species contents, and isotopic data of subsurface fluid samples from the southeastern Tibetan Plateau (SETP), as well as the strain rate data in the study area to (i), constrain the depths of plateau growth dynamics, and (ii) evaluate how the India-Asia convergence dominates the stress distribution in the SETP. This review mainly focuses on the authors' response to the reviewer's comment as requested by the editor. In general, all of the comments from the reviewer have been well responded to and addressed, except for that comment on the statistical test. With the Pearson or Spearman test, both r (or R^2) and p values should be presented. The p -value is an indicator of the statistical significance of the studied correlation. In addition, the reviewer has two minor comments as shown below:

1. Line 191-194: although.....however.... please fix.
2. Calcite precipitation should be explicitly noted in figure 3a.

Tao Wen
Assistant Professor
Department of Earth and Environmental Sciences
Syracuse University

Reviewer #3 (Remarks to the Author):

The revised manuscript has been improved considerably by the authors compared to the original version, which is commendable, and the authors have responded very well to the comment and questions I posed in my original review. There are a few minor points that require clarification/correction, as outlined below. Once these have been addressed, I would recommend publication of the manuscript.

Best wishes,

Wouter Schellart

61 "anomalies", not "anamolies"

95 "data were", not "data was"

142-143 Indeed, there is a trend with increasing $^3\text{He}/^4\text{He}$ with increasing distance from the IACB, but this trench is based on only three data groups (TRF, LTF, XSHF). This limitation should be acknowledged in the text.

235-236 "second invariant of strain rate (equivalent to total strain rate)" The second invariant of the strain rate is not the same as the total strain rate (e.g. it does not contain the first invariant of the strain rate).

257 "compiled", not "complied"

310-311 What do you mean with "the stresses are much more widely distributed" In general, one

speaks of distributed deformation (and localised deformation), but not of distributed stresses.

311 "indicated" is not correct, at most you could say "implied". But then again, strain rate magnitude is determined by both stress and rheology.

319 "undergoing" Do you mean "ongoing"?

337 "strong stresses" What do yo mean? High compressive stresses?

345-346 "the stresses exerted on the interior faults of the SETP are more distributed than those of the boundary fault." Again, what do you mean with more distributed stresses?

369 It would be more accurate to say "the strain rate distribution" rather than "the stress distribution"

Response to Reviewers' Comments

Notes on the point-by-point responses

Reviewers' Comments: Plain text, Calibri font

Authors Responses: *Italic, indented dark red text, Calibri font*

Reviewer #1

I commend the authors for comprehensively addressing all of my concerns and I am now happy for the paper to be published in Nature Communications without further revision.

***Response:** We thank the reviewer for recognition of our work, and for the constructive comments that have improved the geochemical interpretation of our data.*

Reviewer #2

The authors combined the use of helium, and stable carbon and nitrogen species contents, and isotopic data of subsurface fluid samples from the southeastern Tibetan Plateau (SETP), as well as the strain rate data in the study area to (i), constrain the depths of plateau growth dynamics, and (ii) evaluate how the India-Asia convergence dominates the stress distribution in the SETP. This review mainly focuses on the authors' response to the reviewer's comment as requested by the editor. In general, all of the comments from the reviewer have been well responded to and addressed, except for that commend on the statistical test. With the Pearson or Spearman test, both r (or R^2) and p values should be presented. The p -value is an indicator of the statistical significance of the studied correlation. In addition, the reviewer has two minor comments as shown below:

***Response:** We thank the reviewer for the constructive comments, which have improved the geochemical interpretation of our data and the data presentation methods. We have presented the p values in corresponding places (i.e., Lines 150, 267–268, 272–274, and 279–280) of the revised manuscript. In addition, the 95% BCa confidence interval is also shown for the Pearson's r value.*

1. Lines 191–194: although.....however.... please fix.

***Response:** Thanks for suggestion. We have fixed this sentence in the revised manuscript (Lines 199–203).*

2. Calcite precipitation should be explicitly noted in figure 3a.

Response: Thanks for suggestion. We have shown the effect of calcite precipitation in revised version of Fig. 3a following the calcite model presented in Barry et al. (2020, CG, <https://doi.org/10.1016/j.chemgeo.2020.119722>).

Reviewer #3

The revised manuscript has been improved considerably by the authors compared to the original version, which is commendable, and the authors have responded very well to the comment and questions I posed in my original review. There are a few minor points that require clarification/correction, as outlined below. Once these have been addressed, I would recommend publication of the manuscript.

Response: We thank the reviewer for the constructive comments, which have improved the interpretation of geochemical data from the tectonic and geodynamic points of view.

Line 61. "anomalies", not "anamolies".

Response: Thanks. We have corrected this typo (Line 63 of the revised manuscript).

Line 95. "data were", not "data was".

Response: Thanks. We have corrected this typo (Line 97 of the revised manuscript).

Lines 142–143. Indeed, there is a trend with increasing $^3\text{He}/^4\text{He}$ with increasing distance from the IACB, but this trend is based on only three data groups (TRF, LTF, XSHF). This limitation should be acknowledged in the text.

Response: Yes, we agree with the reviewer. We have acknowledged this limitation in the revised manuscript (Lines 152–154).

Lines 235–236. "second invariant of strain rate (equivalent to total strain rate)". The second invariant of the strain rate is not the same as the total strain rate (e.g. it does not contain the first invariant of the strain rate).

Response: Yes, we agree with the reviewer that the second invariant of strain rate is not the same as the total strain rate, because they have different definitions and meanings. The (total) strain rate tensor can be subdivided into isotropic and deviatoric components. Specifically, the isotropic components are normal strain rates that reflect the rates of axial deformation (i.e., extension/compression) across the surface, while the deviatoric components are shear strain rates that reflect the rate of shear deformation. In the case of 2D shear stress and the resultant shear deformation, the total shear strain rate tensor can be described by two invariants. The first invariant of strain rate is used to describe the trace (or direction) of the strain rate tensor. The second invariant of strain rate can

reflect the magnitude of total strain rate (i.e., they are comparable in numerical values).

The Global Strain Rate Model v2.1 is based on horizontal GPS velocities (Kreemer et al., 2014, G-cubed, <https://doi.org/10.1002/2014GC005407>); and it is thus a 2D model. According to the methods presented in Kreemer et al. (2014), the calculated second invariant of strain rates [= $\sqrt{\dot{\epsilon}_{xx}^2 + \dot{\epsilon}_{yy}^2 + 2\dot{\epsilon}_{xy}^2}$] are nearly identical in numerical values with the total strain rates [= $\sqrt{\dot{\epsilon}_1^2 + \dot{\epsilon}_2^2}$]. This is explicitly shown by the plot of strain rate data of the SETP and adjacent region below.

Therefore, we have revised this sentence (Lines 243–244) and will only use total strain rate in the revised manuscript. The second invariant of strain rate will not be mentioned in the revised manuscript, because it is nearly identical in numerical value with the total strain rate.

Line 257. "compiled", not "complied".

Response: Thanks. We have corrected this typo (Line 271 of the revised manuscript).

Lines 310–311. What do you mean with "the stresses are much more widely distributed" In general, one speaks of distributed deformation (and localized deformation), but not of distributed stresses.

Response: Thanks for pointing out this inaccurate use of terminology. We have removed "more distributed stresses" from revised manuscript based on the reviewer's suggestion.

What we meant to say was that the stresses exerted along the interior faults are lower and not so focused within the fault zone compared with the case of the boundary fault. This is consistent with the distributed deformation in the interior of the SETP (Gan et al.,

2007, JGR, <https://doi.org/10.1029/2005JB004120>), and localized deformation along the boundary fault (Zhang et al., 2017, EPSL, <https://doi.org/10.1016/j.epsl.2017.02.025>).

Line 311. "indicated" is not correct, at most you could say "implied". But then again, strain rate magnitude is determined by both stress and rheology.

Response: *Thanks for suggestion. We have changed "indicated" to "implied" (Line 321 of the revised manuscript).*

Yes, we agree with the reviewer that the magnitude of strain rate is determined by both stress and rheology. In this study, we also suggest that the stress plays a fundamental role in strain partitioning across the active faults. Importantly, the boundary fault and interior faults have contrasting total strain rates and strike-slip rates (including geodetic and geological slip rates). These observations (especially slip rate data) indicate different stresses between the boundary and interior faults over geological timescales, consistent with their differences in strain rate. The role of rheology may be less important.

Line 319. "undergoing". Do you mean "ongoing"?

Response: *Yes. We have changed "undergoing" to "ongoing" in the revised manuscript (Line 337).*

Line 337. "strong stresses". What do you mean? High compressive stresses?

Response: *Yes, the Xianshuihe fault is a large-scale strike-slip fault and is characterized by high compressive stresses, which can account for the transpressional deformation since its initiation in mid to late Miocene time. We have revised this sentence (Line 356) based on the reviewer's suggestion.*

Lines 345–346. "the stresses exerted on the interior faults of the SETP are more distributed than those of the boundary fault." Again, what do you mean with more distributed stresses?

Response: *We have avoided any occurrence of "more distributed stresses" in the revised manuscript. We suggest that interior faults of the SETP are controlled by relatively lower stresses than those of the boundary fault; and moreover, such lower stresses are not so focused along the interior faults, in contrast to the case of plateau boundary fault. This interpretation is consistent with our He degassing model and geodetic measurements.*

Line 369. It would be more accurate to say "the strain rate distribution" rather than "the stress distribution".

Response: *Thanks for suggestion. We have changed "the stress distribution" to "the strain rate distribution" in the revised manuscript (Line 388).*